# CHN-CH4: A Gridded (0.1°× $0.1°$) Anthropogenic Methane Emission Inventory of China from 1990 to 2020

Fengxiang Guo[1], Fan Dai[2], Peng Gong[1], Yuyu Zhou[1, 2]

[1]Department of Geography, The University of Hong Kong, Hong Kong, PR China;
[2]Institute for Climate and Carbon Neutrality, The University of Hong Kong, Hong Kong, PR China
*Correspondence to*: Fan Dai (fandai@hku.hk), Yuyu Zhou (yuyuzhou@hku.hk)

**Abstract.** China is the largest emitter of global methane emissions, contributing about 10% to anthropogenic climate change based on existing methane inventories. However, significant uncertainties in these statistics limit the accuracy at both national and sub-national scales. The lack of continuous gridded emissions inventories also constrains the inverse analysis of atmospheric observations. To address these, we present CHN-CH4, a spatially aggregated 0.1°×0.1° anthropogenic methane emission inventory for mainland China from 1990 to 2020 annually. CHN-CH4 offers the country with new temporal coverage and details, by means of national statistical yearbooks and remote sensing products. Over the three decades, mainland China emitted 37.3 [28.1-46.4] Tg of methane annually, with the highest emission occurred in the last decade. But this decade also marked the beginning of a decreasing trend, from 45.0 [32.8- 56.5] Tg in 2010 to 43.4 [31.6-53.4] Tg in 2020. As important priors, CHN-CH4 enables robust comparisons between estimated emissions and atmospheric observations, thereby improving the accuracy of inverse modelling, which is crucial for effective tracking of methane emissions. By providing a reliable and detailed emissions inventory, CHN-CH4 would be a valuable tool in accelerating the global effort to achieve equitable methane emission reduction goals, as well as supporting China's climate policy.

## 1 Introduction

Methane is a potent greenhouse gas, possessing a global warming potential 28 times greater than carbon dioxide over a 100-year period (IPCC, 2013; Peng et al., 2016; Qu et al., 2021). Since 1750, anthropogenic methane emissions have contributed approximately 0.97 $Wm^{-2}$ to radiative forcing, accounting for 32% of the total radiative forcing from long-lived greenhouse gases and about 20% of anthropogenic climate change (IPCC, 2013; Saunois et al., 2020). China, as the largest global emitter, contributed an estimated 16% of global emissions in the 2000s, primarily driven by coal mining, rice cultivation, and livestock production (Kirschke et al., 2013; IEA, 2019). Aligned with the Paris Agreement goals and Global Methane Pledge (European Commission and United States of America, 2021), China has set forth policies to significantly reduce methane emission in its 14th Five-Year-Plan (State Council of China, 2021) and the recent released National Methane Action Plan (Ministry of Ecology and Environment of China, 2023). The rapid and sustained reduction of methane emissions is crucial for achieving collective climate mitigation goals and carbon neutrality, and it offers significant co-benefits for the economy and public health (IPCC, 2006; IPCC, 2019; Maasakkers et al., 2023;

Saunois et al., 2024). Methane emission inventories are essential for identifying priority regions and sectors, thereby supporting the development of effective reduction strategies both regionally and globally. Inverse analysis of atmospheric observations can enhance the robustness of methane emission estimates, but it requires a gridded emissions inventory as prior knowledge (IPCC, 2019; Maasakkers et al., 2023; Sheng et al., 2019). This inventory serves as the basis for interpreting inverse results at specific locations and across various sectors. However, the emissions inventories submitted by each nation to the United Nations Framework Convention on Climate Change (UNFCCC) often lack accurate spatial information. This lack of detailed spatial data creates knowledge gaps in regulating methane emission sources, formulating reduction policies, and fostering international cooperation. Meanwhile, the quality of the inventory significantly affects the final methane estimates (Maasakkers et al., 2023). For China, most of previous studies related to inverse modelling have used the global gridded inventory from the Emissions Database for Global Atmospheric Research (EDGAR) (Lin et al., 2021; Chen et al., 2022; Shen et al., 2023). While EDGAR enables spatial comparisons with observations, aggregated national inventories exhibit inconsistent trends and estimates at the local scale (Sheng et al., 2017; Behrendt et al., 2025). A 30%~40% overestimation for the period 1980-2008 was identified by the work from Peng et al. (2016). Lin et al. (2021) systematically compared anthropogenic methane emissions based on bottom-up inventories. They found an over 11 Tg/year overestimation of EDGAR for period 1990-2010 in China, by comparison to PKU-CH4, another widely applied methane emission inventory. Another weakness in the literature is the lack of downloadable continuous emission inventories at the grid level, limited by proxy data collection, particularly after 1990 (Zhang and Chen, 2014; Peng et al., 2016; Lin et al., 2021; Gong and Shi, 2021), which is the baseline year used by the UNFCCC and the Kyoto Protocol. Releasing national gridded methane emission inventories for the period after 1990 is necessary to maintain consistency and comparability with international emission reduction commitments. The recent development of satellite remote sensing technology makes this possible, as it provides high-resolution proxy data for estimating methane emissions over the years.

In this study, we present a spatially aggregated 0.1°× 0.1° anthropogenic methane emission inventory for mainland China (CHN-CH4) annually, utilizing satellite products, national statistical databases and existing other proxy datasets. This inventory allows for the evaluation of annual trends over the period 1990-2020 across three dominant sectors: agriculture, energy, and waste. It further delineates emissions into eight major source sectors: 1) rice cultivation, 2) livestock, 3) biomass and biofuel burning, 4) coal exploitation, 5) oil and natural gas (NG) systems, 6) fossil fuel consumption, 7) landfills, and 8) wastewater. To ensure maximum comparability, we incorporated recent proxy data, preferably long-term remote sensing data. We also considered emission factors based on measured statistical data specifically tailored for mainland China. Note that this inventory includes only annual anthropogenic methane emissions and excludes the emissions from natural sources.

## 2 Method

The methane emissions from three dominant sectors—agriculture (including livestock, rice cultivation, biomass & biofuel burning), energy (including coal exploitation, oil and NG systems, and fossil fuel

combustion), and waste management (including landfills and wastewater management)—are considered in this dataset. Figure. 1 illustrates the key steps for estimating the gridded methane emissions based on satellite observation products, multiple statistical yearbooks, and existing emissions datasets from other sources.

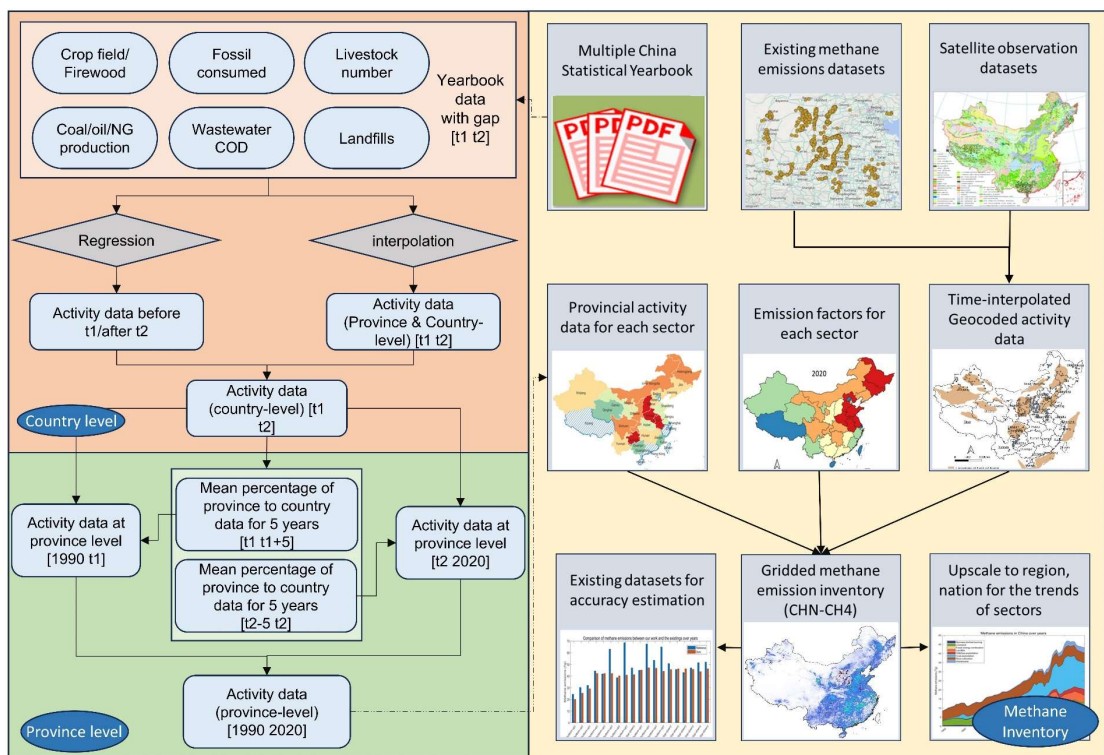

**Figure 1. Flowchart for mapping anthropogenic methane emissions in China during period 1990-2020. The left subfigure displayed how to handle with the missing data from the statistical yearbooks at nation and province level. t1 and t2 represents the earliest and most recent available year that we can get the data from statistical yearbooks. The right subfigure displayed the steps for the methane estimates and comparison with existing inventories.**

The methods outlined in the 2006 IPCC Guidelines for National Greenhouse Gas (IPCC, 2006) and the 2019 Refinement to the 2006 IPCC Guidelines for National Greenhouse Gas Inventories (IPCC, 2019) are used to estimate methane emissions for these eight sectors. The annual methane emissions are calculated by

$$E(S,t) = \sum_C AD_{S,G,C}(t) * EF_{S,R,C}(t) * \left(1 - CF_{S,R,C}(t)\right), \tag{1}$$

where $E(S,t)$ represents the total methane emissions from sector $S$ in the inventory year $t$. The parameter $AD_{S,G,C}(t)$ refer to the activity data in the year $t$ for sector S, grid G belonging to provinces $R$ and condition C. Table. 1 provided detailed information about the activity data for each sector. $EF$ is the emissions factor and we used provincial emission factors to calculate the gridded emissions corresponding to each province. $CF$ is the correction factor, which indicates the fraction of methane utilized or oxidized without being released to atmosphere.

**Table 1. Data information used for CHN-CH4.**

| Sectors | Content | Proxy data | Information |
|---|---|---|---|
| Agriculture | Rice cultivation | CCD-Rice | 30 m × 30 m, 1990-2016[a] |
| | | ChinaCP | 500 m × 500 m, 2015-2021[b] |
| | Livestock | Livestock distributions | 0.1° × 0.1°, 2010, 2015, 2020[c] |
| | | Livestock population | Provincial, 1990-2020[d] |
| | Biomass & biofuel burning | Gridded population | 30 arc-seconds, 1990-2020[e, f] |
| | | GISA (Impervious surface datasets) | 30 m × 30 m, 1972-2019[g] |
| | | Amounts of crop areas, yields | Provincial, 1990-2020[d] |
| | | Amounts of firewood consumption | Provincial, 1991-2007[h] |
| Energy | Coal exploitation | Coal production | Provincial, 1990-2020[h] |
| | | EDGARv8.0 | 0.1° × 0.1°, 1970-2022[i] |
| | | Climate Trace | Point, 2015-2022[j] |
| | Oil and natural gas systems | Oil production | Provincial, 1990-2020[h] |
| | | Natural gas production | Provincial, 1990-2020[h] |
| | | EDGARv8.0 | 0.1° × 0.1°, 1970-2022[i] |
| | Fossil fuel combustion | Gridded GDP | 30 arc-seconds, 1990-2020[k, l] |
| | | Coal consumption | Provincial, 1990-2020[h] |
| | | Oil consumption | Provincial, 1990-2020[h] |
| | | Natural gas consumption | Provincial, 1990-2020[h] |
| Waste management | Landfills | Amounts of MSW landfills | Provincial, 2003-2020[m, n] |
| | | Gridded population | 30 arc-seconds, 1990-2020[e, f] |
| | Wastewater management | Gridded population | 30 arc-seconds, 1990-2020[e, f] |
| | | COD amounts of industrial wastewater | Provincial, 1993-2020[m, n] |
| | | COD amounts of domestic sewage | Provincial, 1993-2020[m, n] |

Notes: [a] Shen et al., 2025; [b] Qiu et al., 2022; [c] FAO (http://www.fao.org/livestock-systems/en/); [d] China Agricultural Statistical Yearbook; [e] Wang and Wang, (2022a); [f] WorldPop (https://www.worldpop.org/); [g] Huang et al., 2021; [h] China Energy Statistical Yearbook; [i] EDGAR v8.0 (https://edgar.jrc.ec.europa.eu/); [j] Climate Trace (https://climatetrace.org/); [k] Wang and Wang, (2022b); [l] Zhao et al., 2017; [m] China Statistical Yearbook; [n] China Environmental Statistical Yearbook. Date in the information column represents the start and end time of province-level that we can collect from various national statistical yearbooks. And the data missing phenomenon exists between them.

**2.1 Methods for agriculture sector**

**2.1.1 Rice Cultivation**

The methane emissions from rice cultivation sector are estimated following the methodology of Fu and Yu (2010) using the following equation:

$$E(t) = \sum_i A_{G,i}(t) * EF_{R,i} * p_i, \tag{2}$$

where $A$ is the cultivated rice area, $i$ is the rice season division (early, middle and late) and $p$ is the rice growing period. In Northern China, only single-harvest rice is cultivated, while in Southern China, double-harvest early rice, single-harvest rice, and double-harvest late rice are grown. In this project, we define single-harvest rice as middle rice. The early and late rice are assumed to be planted in the same area. The growing periods for early, middle, and late rice are 77 days, 110–130 days, and 93 days,

respectively (Peng et al., 2016; Yan et al., 2003). Table. 2 lists the emission factors ($EF_s$) for each rice season by region, tailored to the Chinese context. These $EF_s$ are based on 204 season-treatment measurements across 23 different sites. Due to the limitations of existing satellite products, which do not cover the entire period from 1990 to 2020, we used two datasets for gridded rice cultivation areas annually: CCD-Rice for the period 1990-2016 (Shen et al., 2025) and ChinaCP for the period 2017-2020 (Qiu et al., 2022). The CCD-Rice dataset was derived from Landsat Collection 2 Level-2 Science Products at 30 m spatial resolution, with provincial-level distribution maps showing an average overall accuracy of 89.61% and strong coefficients of determination ($R^2$ = 0.85 for single-season rice and 0.78 for double-season rice). The ChinaCP dataset was developed from MODIS imagery at 500 m resolution using phenology-based mapping algorithms. The validation against ground truth data revealed an overall accuracy of 89%, with excellent agreement to statistical data ($R^2 \geqslant 0.89$). The accuracy is further applied to evaluate the uncertainty caused by rice paddy area. These datasets were then resampled into 0.1° by 0.1° gridded maps. The uncertainty of methane emissions from rice cultivation is derived from the range of $EF_s$ (Peng et al., 2016; Yan et al., 2003).

**Table 2. $EF_s$ of rice cultivation, underground mining and the fraction of burning as biofuels or in the open fields.**

| | $EF_s$ of rice cultivation ($kg/ha^2/day$) | | | Fraction of burning crop residues | | $EF_s$ of underground mining ($m^3 ton^{-1}$) |
|---|---|---|---|---|---|---|
| | early | middle | late | Biomass fuels | Open burning | |
| Beijing | 0 | 1.26 | 0 | 0.7 | 0.05 | 0.739 |
| Tianjin | 0 | 1.08 | 0 | 0.7 | 0.05 | 0 |
| Hebei | 0 | 1.46 | 0 | 0.4 | 0.1 | 3.997 |
| Shanxi | 0 | 0.63 | 0 | 0.45 | 0.1 | 6.532 |
| Inner Mongolia | 0 | 0.85 | 0 | 0.4 | 0.05 | 0.899 |
| Liaoning | 0 | 0.88 | 0 | 0.55 | 0.1 | 11.402 |
| Jilin | 0 | 0.53 | 0 | 0.3 | 0.2 | 5.926 |
| Heilongjiang | 0 | 0.79 | 0 | 0.55 | 0.2 | 17.275 |
| Shanghai | 0 | 1.46 | 2.75 | 0.2 | 0.2 | 0 |
| Jiangsu | 0 | 1.89 | 2.76 | 0.8 | 0.05 | 4.228 |
| Zhejiang | 1.69 | 1.69 | 3.45 | 0.45 | 0.2 | 0 |
| Anhui | 1.97 | 1.97 | 2.76 | 0.8 | 0.05 | 20.477 |
| Fujian | 0.91 | 0.91 | 5.26 | 0.3 | 0.2 | 0.739 |
| Jiangxi | 1.82 | 1.82 | 4.58 | 0.45 | 0.1 | 11.014 |
| Shandong | 0 | 2 | 0 | 0.45 | 0.1 | 1.168 |
| Henan | 0 | 1.7 | 0 | 0.3 | 0.1 | 13.062 |
| Hubei | 2.06 | 2.06 | 3.9 | 0.7 | 0.1 | 7.833 |
| Hunan | 1.73 | 1.73 | 3.41 | 0.4 | 0.1 | 9.953 |
| Guangdong | 1.77 | 1.77 | 5.16 | 0.55 | 0.2 | 0 |
| Guangxi | 1.46 | 1.46 | 4.91 | 0.45 | 0.1 | 0.707 |
| Hainan | 1.58 | 1.58 | 4.94 | 0.45 | 0.1 | 0 |
| Chongqing | 0.77 | 0.77 | 1.85 | 0.7 | 0.1 | 33.685 |
| Sichuan | 0.77 | 0.77 | 1.85 | 0.45 | 0.1 | 12.674 |
| Guizhou | 0.6 | 0.6 | 2.1 | 0.4 | 0.1 | 24.122 |

| Yunnan | 0.28 | 0.28 | 0.76 | 0.2 | 0.1 | 8.787 |
|---|---|---|---|---|---|---|
| Xizang | 0 | 0.65 | 0 | 0.2 | 0.05 | 0 |
| Shaanxi | 0 | 1.19 | 0 | 0.45 | 0.1 | 2.511 |
| Gansu | 0 | 0.65 | 0 | 0.55 | 0.05 | 1.916 |
| Qinghai | 0 | 0 | 0 | 0.8 | 0.05 | 0.739 |
| Ningxia | 0 | 0.7 | 0 | 0.45 | 0.05 | 8.985 |
| Xinjiang | 0 | 1 | 0 | 0.2 | 0.05 | 3.141 |

**2.1.2 Livestock**

The methane emissions from livestock sector include the activities of enteric fermentation and manure management. Enteric fermentation is the process by which microbes in the digestive system of ruminant animals produce methane. Methane is also emitted from the collection, storage, treatment, and disposal of animal manure. The emissions from these activities are calculated using the following formula:

$$E(t) = \sum_i N_{G,i}(t) * EF_{R,i}, \tag{3}$$

where $i$ is the category, and $N$ is the number of ruminant and non-ruminant animals at grid $G$. Table. 3 lists the considered animals and the widely applied values of emission factors for two sources: enteric fermentation and manure management. We selected the emission factors from Zhou et al. (2007), which have been widely used to calculate methane emissions from livestock in China (Zhou et al., 2007; Fu and Yu, 2010; Zhang and Chen, 2010, 2014). For manure management from swine, we used the value of 3.05 $kg/head/year$ following the work of CCCCS (2000) and Zhang and Chen (2014). For the uncertainty analysis, we used the maximum and minimum values of the emission factors for each category listed in Table. 3 as the range.

**Table 3. $EF_s$ of enteric fermentation and manure management for livestock.**

| | Enteric fermentation ($kg/head/year$) | | | | Manure management ($kg/head/year$) | | | | | | | |
|---|---|---|---|---|---|---|---|---|---|---|---|---|
| | [a] | [b] | [c] | [d] | [e] | | | | [c] | | | [d] |
| | | | | | cool | temp | warm | average | cool | temp | warm | |
| Non-dairy cattle | 44 | | 47 | 54.21 | 0.65 | 0.92 | 1.97 | 0.77 | 1 | 1 | 1 | 0.92 |
| Dairy cattle | 44 | | 61 | 65.25 | 7.65 | 16.36 | 26.17 | 8.87 | 9-12 | 13-26 | 28-31 | 8.95 |
| Buffalo | 50 | 56.3 | 55 | 72.92 | 0.92 | 1.07 | 2.35 | 1.07 | 1 | 2 | 2 | 1.8 |
| Sheep | 5 | 5.6 | 5 | 5.34 | 0.1 | 0.15 | 0.2 | 0.1 | 0.1 | 0.15 | 0.2 | 0.1 |
| Goats | 5 | 5.4 | 5 | 4.62 | 0.11 | 0.17 | 0.22 | 0.13 | 0.11 | 0.17 | 0.22 | 0.13 |
| Swine | 1 | 1 | 1 | 1 | 1.26 | 3.74 | 7.09 | 3.05 | 2 | 3-6 | 6-7 | 1.53 |
| Poultry | | | | | 0.012 | 0.018 | 0.023 | 0.016 | 0.01 | 0.02 | 0.02 | 0.015 |

Notes: [a] Khalil et al., (1993); [b] Yamaji et al., 2003; [c] IPCC (2006); [d] Zhou et al., (2007); [e] CCCCS (2000).

For the activity data, we used the gridded map from the Food and Agriculture Organization of the United Nations (FAO) that provides the global livestock density (GLW) dataset based on the work of Gilbert et al. (2018) and Gilbert et al. (2022). However, the official website only released data for the years 2010, 2015, and 2020, which do not cover the entire target period of 1990-2020. For the other years without gridded data, we attributed the province-level data to each grid based on the spatial proportion belonging to the same province in the year 2010. We collected the province-level annual census data of domestic

livestock for each livestock category from China Agricultural Statistical Yearbook. The comparable results show good consistency with the reference grid data for 2015 and 2020 from FAO (Fig. S1 in the supplementary file). Notably, since seasonal births and slaughters change the population of livestock, we combined the slaughtered population and live population together as the Net Annual Population Average (*NAPA*) at the end of the year. We then calculated the number of each livestock category, following the Guidelines of IPCC (2006):

$$N_i = Days_{alive, i} * \frac{NAPA_i}{365}, \tag{4}$$

The average life spans in one year are 12 months for dairy cattle, 10 months for non-dairy cattle and buffalo, 7 months for sheep and goats, 6 months for swine and 2 months for poultry.

### 2.1.3 Biomass and Biofuel burning

The emissions from this sector come from the straw in rural households and the burning of firewood. The former mainly refers to crop residues used as biofuels in houses and those burnt in open fields, based on the work of (Fu and Yu, 2010)Tian et al. (2010), which is calculated by

$$E(t) = EF * \sum_i R_i * N_{G,i}(t) * F * \theta, \tag{5}$$

where $R_i$ is the straw/grain ratio for the crop $i$: rice (0.623), wheat (1.366), corn (2), other grains (1), bean (1.5), tuber (0.5), cotton (3), fiber crop (1.7), sugar crop (0.1), and oil-bearing crop (2) (Fu and Yu, 2010). $F$ is the fraction of crop residues used as biofuels in houses or burnt in open fields (Table. 2), while $\theta$ is the corresponding burning efficiency: 100% for biofuels and 88.9% for fires in open fields (Peng et al., 2016; Tian et al., 2010; Zhang et al., 2008). $N_{G,i}$ represents the annual yields at grid $G$ for the crop i. Since straw is mainly used by rural residents, we generated the proxy data at the grid level by attributing the province-level data to the spatial distribution of rural populations. Province-level data comes from the China Agricultural Statistical Yearbook, while the rural populations are mapped by masking the grid dataset of China's historical population (Wang and Wang, 2022a; WorldPop) with the impervious surface datasets GISA (Huang et al., 2021).

For the estimation of methane emissions from firewood burning, we collected the province-level data on firewood consumed as biofuels from the China Energy Statistical Yearbook and then attributed this data to the gridded rural population data. Since no firewood data are available after 2007, we performed a linear regression with the rural population to fill in the missing data. The $EF_s$ of methane emissions from biomass and biofuel burning come from the existing literature (Zhang et al., 2000, 2008; Fu and Yu, 2010; Peng et al., 2016). We used the $EF_s$ of crop residues for biomass fuel $3.62 \pm 2.20\ kgCH_4t^{-1}$ and fire in open fields $3.89 \pm 2.20\ kgCH_4t^{-1}$, respectively. For the firewood consumption as biofuels, the $EF_s$ $2.77 \pm 1.80\ kgCH_4t^{-1}$ was adopted.

### 2.2 Methods for energy sector

### 2.2.1 Coal exploitation

Methane emissions from coal exploitation include fugitive methane from underground, surface, and post-

mining activities (IPCC, 2006). To estimate the emissions from this sector, it is crucial to determine the gridded distribution of coal exploitation amounts. We combined the methane emission map for the period 1990-2020 from the EDGAR v8.0 Fuel Exploitation datasets (European Commission & Joint Research Centre, 2023) with province-level coal production data from the China Energy Statistical Yearbook (1990–2020) and the China Statistical Yearbook (1990–2020). First, we separated the locations of coal exploitation from oil and natural gas systems in the Fuel Exploitation sector of EDGAR v8.0 using the following criteria:

- Point-source and line-source methane emissions were distinguished by setting a threshold of 50.
- Point-source methane emission locations outside the tight oil basins in China were classified as coal mining locations (Fig. 2).
- For point-source methane emission locations within the tight oil basins, we used the Climate TRACE (CT) oil and gas dataset to identify and distinguish coal mining locations (Fig. 2).
- The remaining point-source locations were classified as oil and natural gas locations.

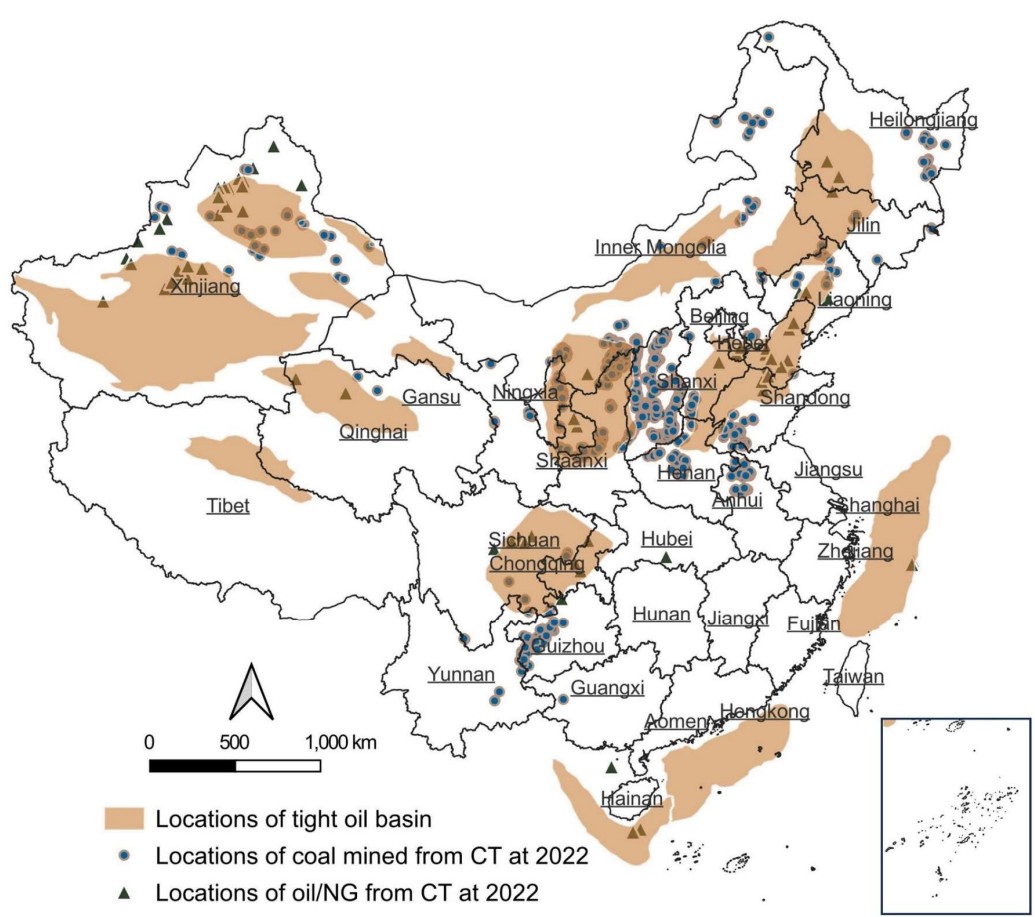

**Figure 2. Spatial distributions of tight oil basin, coal mines, and oil/NG from Climate Trace at 2022.**

Next, we calculated the spatial proportion of methane emissions for each coal mining location relative to the total methane emissions for the corresponding province, based on the Fuel Exploitation methane emission map from EDGAR v8.0. Finally, we attributed the province-level coal production volumes to each corresponding coal mining location based on the above proportion.

For methane emissions from underground mines, Zhu et al. (2017) developed province-level methane

emission factors ($EF_s$) for coal mining in China for the period 2005–2010. These $EF_s$ were derived from data analysis of coal production and corresponding methane emissions from 787 coal mines across 25 provinces with diverse geological and operational conditions. We used the mean values to estimate methane emissions from underground mining, while the minimum and maximum values were applied to analyze uncertainties (Table. 2).

Methane emissions from surface mining are generally much lower than those from underground mining. On average, only about 5% of coal was mined from surface mines at the national scale (Peng et al., 2016; Zhu et al., 2017). Due to the limited availability of methane emission measurements for surface mining, we adopted the default $EF$ of 2.5 $m^{-3}$ per tonne of coal mined, as recommended by IPCC (2006).

Currently, there are no direct measurements available for methane emissions from post-mining activities. Following Zheng et al. (2005) and Zhu et al. (2017), we assumed that 12% of the total methane emissions from coal mining are associated with emissions during the subsequent handling, processing, and transportation of coal. In China, the fraction of methane collected and utilized from coal mining operations ($CF_{S,R,C}$) has increased with economic growth and improvements in coal safety (NDRC, 2014; Peng et al., 2016). Based on the findings of Peng et al. (2016) and Zhu et al. (2017), the recovery fraction increased linearly from 3.59% in 1994 to 9.4% in 2010. For our study, we used a recovery fraction of 3.59% for the period before 1994 and 9.4% after 2010. To account for uncertainty, we adopted a range of recovery fractions (3.59%–5.21%) based on the studies by Zheng et al. (2005) and Peng et al. (2016).

**2.2.2 Oil and natural gas systems**

This sector encompasses methane emissions from venting, flaring, exploration, production, upgrading, transport, refining/processing, transmission, storage, and distribution networks. In IPCC subcategory 1B2 (Fugitive Emissions from Oil and Gas), these emissions are processed as an aggregated source, following the methodologies from Schwietzke et al. (2014) and Peng et al. (2016). For the gridded emissions, province-level annual crude oil and natural gas production data from the China Statistical Yearbook was distributed to each potential oil and natural gas location identified in Section 2.2.1. For methane emissions from oil systems, we applied the $EF_s$ of 0.077 $ktCH_4PJ^{-1}$, with an uncertainty range of 0.058–0.190 $ktCH_4PJ^{-1}$. For the fugitive CH4 from natural gas systems, we used the fugitive emission rates (FERs) provided by Peng et al. (2016) and Schwietzke et al. (2014). These rates assume a linear decrease in FER from 4.6% (equivalent to 0.81 $ktCH_4PJ^{-1}$) in 1980 to 2.0% (0.35 $ktCH_4PJ^{-1}$) in 2010, which aligns with a FER of 1.9% reported for OECD countries. Uncertainty was assessed using two scenarios: 1) A lower FER scenario, where the FER in China decreased from 3.2% in 1990 to 1.4% in 2020; and 2) A higher FER scenario, where the FER decreased from 5.4% in 1990 to 4.6% in 2020. Finally, we combined these emissions with methane emissions from line-source emissions in the Fuel Exploitation sector of EDGAR to obtain the final gridded emissions.

**2.2.3 Fossil fuel consumed**

For gridded emissions in this sector, we collected province-level fossil fuel combustion data from the China Energy Statistical Yearbook for the period 1990–2020. This data was spatially distributed using annual gridded GDP with a 10 km resolution (Wang and Wang, 2022b; Zhao et al., 2017). We adopted

the $EF_s$ from IPCC for methane estimation, 1 $kgCH_4TJ^{-1}$ for coal combustion, 3 $kgCH_4TJ^{-1}$ for oil combustion and 1 $kgCH_4TJ^{-1}$ for natural gas combustion (IPCC, 2006). For uncertainty analysis, we applied a value of 60% of the $EF_s$ for this sector, based on the work of Peng et al. (2016).

### 2.3 Methods for waste sector

### 2.3.1 Landfills

Municipal solid waste (MSW) disposal in China is predominantly handled through landfilling, due to its cost-effectiveness and flexibility in managing varying waste quantities and types. Methane emissions from landfills can be estimated using the first-order decay method according to IPCC guidelines:

$$E(t) = (1 - e^{-k}) * \sum_x e^{-k*(T_L - x)} * MSW_L * MCF_T * F_T * DOC * DOC_d * f * (1 - O_f) * 16/12, \quad (6)$$

where k is reaction constant and $T_L$ is decay lifetime period, which are 0.3 and 4.6 years based on national inventory (NDRC, 2014). $MSW_L$ is total amount of MSW treated by landfills at grid level. $MCF_T$ represents the methane correction factor from three types of landfills: 1.0 for managed anaerobic landfills (Type_A), 0.8 for deep unmanaged landfills exceeding 5 m (Type_B), 0.4 for deep unmanaged landfills within 5 m (Type_C) (IPCC, 2006; NDRC, 2014). $F_T$ is the fraction of $MSW_L$ assigned to each landfill type by province, as shown in Table. 4. $DOC$ is the fraction of degradable organic carbon in $MSW$, and $DOC_d$ is the fraction of DOC that can be decomposed, $f$ is the fraction of methane in landfill gas, and $O_f$ is the oxidation factor, set to 0.1. In this study, we adopted a $DOC$ of 6.5%, $DOC_d$ of 0.6, and $f$ of 0.5 (Ma and Gao, 2011; Zhang and Chen, 2014; Peng et al., 2016). For uncertainty analysis, we assumed maximum methane emissions with $DOC_d = 0.6$ and $f = 0.6$ and the minimum emissions with $DOC_d = 0.5$ and $f = 0.4$.

**Table 4. Classification of MSW disposal sites and average influent/effluent BOD$_5$/COD ratio.**

| | Classification of MSW disposal sites (Du, 2006) | | | Average influent BOD$_5$/COD ratio | Average effluent BOD$_5$/COD ratio |
|---|---|---|---|---|---|
| | Type_A | Type_B | Type_C | | |
| Beijing | 49.2 | 38.1 | 12.7 | 0.46 | 0.28 |
| Tianjin | 54.2 | 34.4 | 11.4 | 0.46 | 0.28 |
| Hebei | 41.8 | 43.7 | 14.5 | 0.46 | 0.28 |
| Shanxi | 35.8 | 48.2 | 16 | 0.46 | 0.28 |
| Inner Mongolia | 25.6 | 55.8 | 18.6 | 0.46 | 0.28 |
| Liaoning | 23.6 | 57.3 | 19.1 | 0.45 | 0.31 |
| Jilin | 17.4 | 62 | 20.6 | 0.45 | 0.31 |
| Heilongjiang | 26.3 | 55.3 | 18.4 | 0.45 | 0.31 |
| Shanghai | 0.9 | 74.3 | 24.8 | 0.42 | 0.26 |
| Jiangsu | 82.1 | 13.4 | 4.5 | 0.42 | 0.26 |
| Zhejiang | 33.7 | 49.7 | 16.6 | 0.42 | 0.26 |
| Anhui | 34.5 | 49.1 | 16.4 | 0.42 | 0.26 |
| Fujian | 36.8 | 47.4 | 15.8 | 0.42 | 0.26 |
| Jiangxi | 24.3 | 56.8 | 18.9 | 0.42 | 0.26 |
| Shandong | 49.5 | 37.9 | 12.6 | 0.42 | 0.26 |
| Henan | 46.5 | 40.1 | 13.4 | 0.48 | 0.34 |
| Hubei | 32.8 | 50.4 | 16.8 | 0.48 | 0.34 |

| Hunan | 62.1 | 28.4 | 9.5 | 0.48 | 0.34 |
| Guangdong | 61.8 | 28.6 | 9.6 | 0.49 | 0.32 |
| Guangxi | 27.8 | 54.1 | 18.1 | 0.49 | 0.32 |
| Hainan | 33.7 | 49.7 | 16.6 | 0.49 | 0.32 |
| Chongqing | 70.2 | 22.3 | 7.5 | 0.51 | 0.31 |
| Sichuan | 46.4 | 40.2 | 13.4 | 0.51 | 0.31 |
| Guizhou | 5.7 | 70.7 | 23.6 | 0.51 | 0.31 |
| Yunnan | 18.9 | 60.8 | 20.3 | 0.51 | 0.31 |
| Xizang | 0 | 75 | 25 | 0.51 | 0.31 |
| Shaanxi | 0 | 75 | 25 | 0.44 | 0.29 |
| Gansu | 25.3 | 56 | 18.7 | 0.44 | 0.29 |
| Qinghai | 58.8 | 30.9 | 10.3 | 0.44 | 0.29 |
| Ningxia | 24.5 | 56.6 | 18.9 | 0.44 | 0.29 |
| Xinjiang | 0 | 75 | 25 | 0.44 | 0.29 |

For the activity data, province-level MSW treated by landfills was collected from China Environmental Statistical Yearbook and China Statistical Yearbook, although data is only available from 2003 onward. Following Peng et al. (2016), we identified a linear association between national $MSW_L$ and GDP (Fig. S2 with $R^2$ over 0.960). We firstly applied the association to estimate 1990-2002 landfill totals and then attributed the national $MSW_L$ to corresponding province using their 2003-2008 average contribution to national landfill amounts. Finally, we downscaled the filled province-level data for 1990-2020 to each grid using the spatial distribution of GDP.

**2.3.2 Wastewater**

Methane emissions from wastewater management arise from industrial wastewater and domestic sewage. For industrial wastewater, the primary factor influencing methane emissions is the degradable organic fraction, commonly represented by chemical oxygen demand (COD). Methane emissions from industrial wastewater treated by wastewater treatment plants (WTPs) and directly discharged into water bodies are calculated as:

$$E(t) = COD_G(t) * EF_{industry} * MCF_{industry}, \tag{7}$$

where $MCF_{industry}$ is the fraction of COD in wastewater treated anaerobically, taken as the correction factor. $EF_{industry}$ is the emission factor for $COD$ removed, set at 0.25 $gCH_4/gCOD$ removed (IPCC, 2006). For $COD$ removed through WTPs, we used an $MCF$ of 0.458 (Ma and Gao, 2011), while an $MCF$ of 0.1 (IPCC, 2006) was adopted for $COD$ directly discharged into waterbodies.

Methane emissions from domestic sewage are typically estimated using biochemical oxygen demand ($BOD_5$) with the equation:

$$E(t) = BOD_5(t) * EF_{domestic} * MCF_{domestic}, \tag{8}$$

Regional $BOD_5$ data in China is not directly available. Instead, we calculated organic loading removed based on COD content in domestic sewage and regional $BOD_5/COD$ ratios. Table. 4 provides these ratios, representing average influent and effluent $BOD_5/COD$ ratios from municipal WTPs (Ma and Gao, 2011; Song, 2011). The MCF for $BOD_5$ removed through WTPs was set at 0.165 (Ma and Gao, 2011), and for

$BOD_5$ discharged into water bodies, we used 0.1 (IPCC, 2006; NDRC, 2014). For uncertainty, we assumed the maximum emissions with $MCF$ = 0.3 for domestic sewage and $MCF$ = 0.5 for industrial wastewater treated by WTPs, and minimum emissions with $MCF$ = 0.1 for domestic sewage and $MCF$ = 0.2 for industrial wastewater treated by WTPs (Ma et al., 2015; Peng et al., 2016).

For the activity data, we collected the province-level and nation-level $COD$ amounts of industrial wastewater and domestic sewage disposed by WTPs or discharged into waterbodies directly from the China Environmental Statistical Yearbook and China Statistical Yearbook. The results in Fig. S3 showed that $COD$ treated by WTPs for both industrial wastewater and domestic sewage increases with GDP ($R^2$ over 0.930), while the fraction of industrial $COD$ directly discharged into water bodies decreases with

GDP, reflecting technological advancements and effective environmental policies. Additionally, $COD$ discharge from domestic sewage increases with population growth ($R^2$ over 0.605). To fill the missing national-level data, we interpolated based on these relationships, distributing values to provinces using their average contribution over the nearest five years. We downscaled the $COD$ amounts treated by WTPs for industrial wastewater and domestic sewage to grids using GDP distribution, and we distributed $COD$

amounts from domestic sewage discharged directly into water bodies based on population distribution.

## 3 Results and Discussions

### 3.1 Evaluation of CHN-CH4 performance with the references

The emissions from CHN-CH4 were comprehensively evaluated against reference inventories at pixel, sectoral, regional and national scales to ensure robust comparisons. Spatial validation against EDGAR

v8 and PKU-CH4 v2 datasets for the years 2000, 2009, and 2019 demonstrates strong agreement with CHN-CH4 along the 1:1 line (Fig. 3). The higher $R^2$ between CHN-CH4 and EDGAR indicate greater spatial consistency compared to PKU-CH4. Notable regional differences persist, with CHN-CH4 showing lower emission estimates than both EDGAR and PKU-CH4 in energy-intensive provinces (e.g., Shanxi and Sichuan) and major rice-growing regions (e.g., Hunan and Jiangxi). Particularly, the

deviations from EDGAR are more pronounced than those from PKU-CH4.

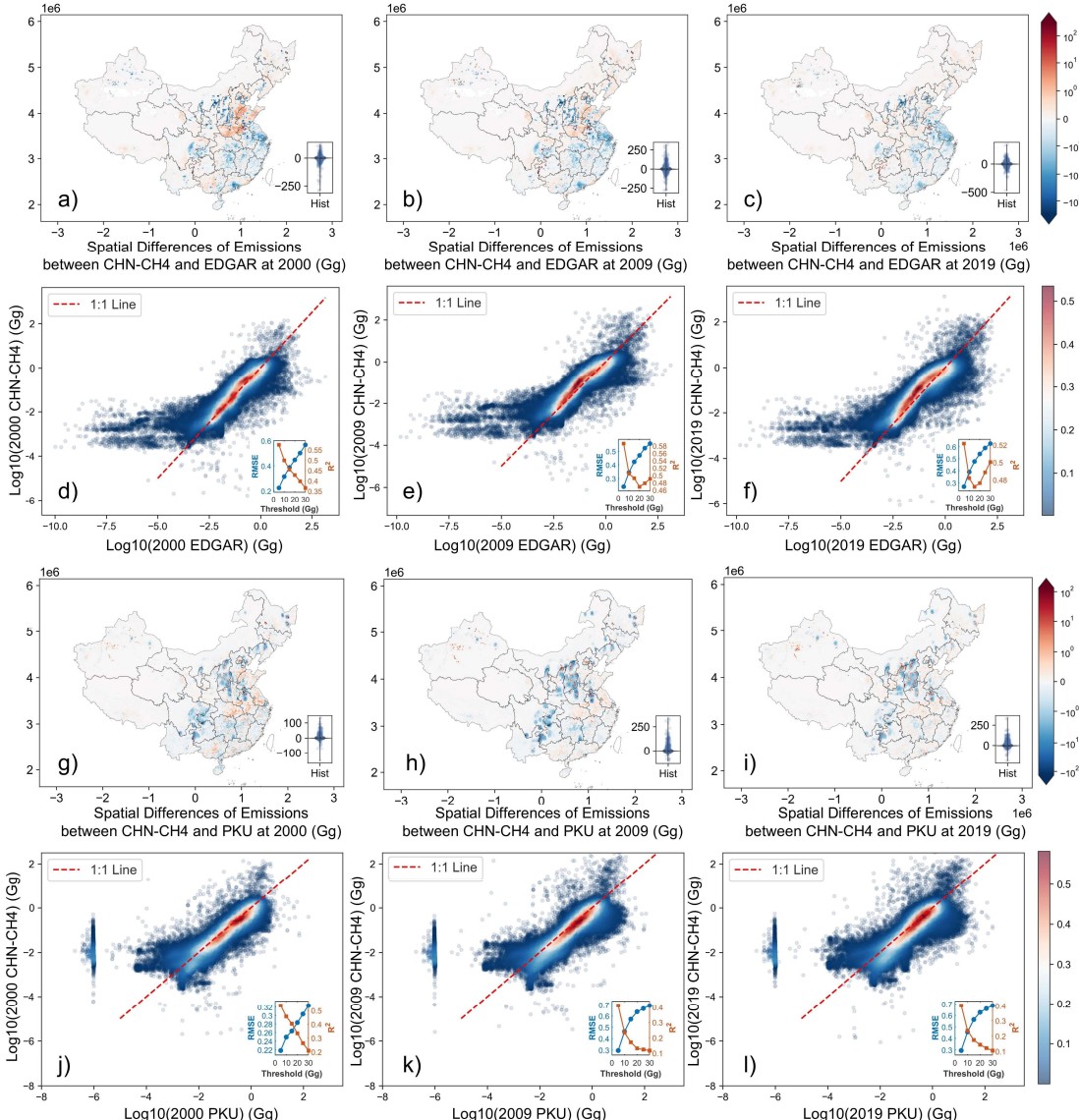

**Figure 3. Pixel-level comparisons between CHN-CH4 and EDGAR/PKU-CH4. a-f) represent the spatial differences between CHN-CH4 and EDGAR v8, while g-l) represent the comparison between CHN-CH4 and PKU-CH4 v2 at year 2000, 2009, and 2019. 'Hist' in each spatial map represents the histogram of the differences between CHN-CH4 and EDGAR/PKU-CH4, with the unit Gg. The bottom-right subfigure in each log-log plot presents threshold-dependent performance metrics, demonstrating how *RMSE* and *$R^2$* vary when excluding grid cells over specific emission thresholds.**

Further comparison with the Global Fuel Exploitation Inventory (GFEI) across energy sectors shows well spatial agreement for oil/NG systems but larger discrepancies in coal mining (*RMSE* > 39.4 Gg, MAE >24.5 Gg) (Fig. 4). In contrast, the oil/NG sector demonstrates much closer alignment, as evidenced by the log-log plot (Fig. 4c and 4f). Compared to GFEI and PKU-CH4, CHN-CH4 exhibits stronger spatial consistency with EDGAR, as evidenced by a higher concentration of points along the 1:1 line. The spatial discrepancies between GFEI and EDGAR primarily arise from differences in their underlying data sources for fuel exploitation; EDGAR's coal and oil/gas system locations were adopted in this study. Despite these variations, the overall alignment supports the reliability of CHN-CH4 in

identifying key sectoral and regional emission patterns.

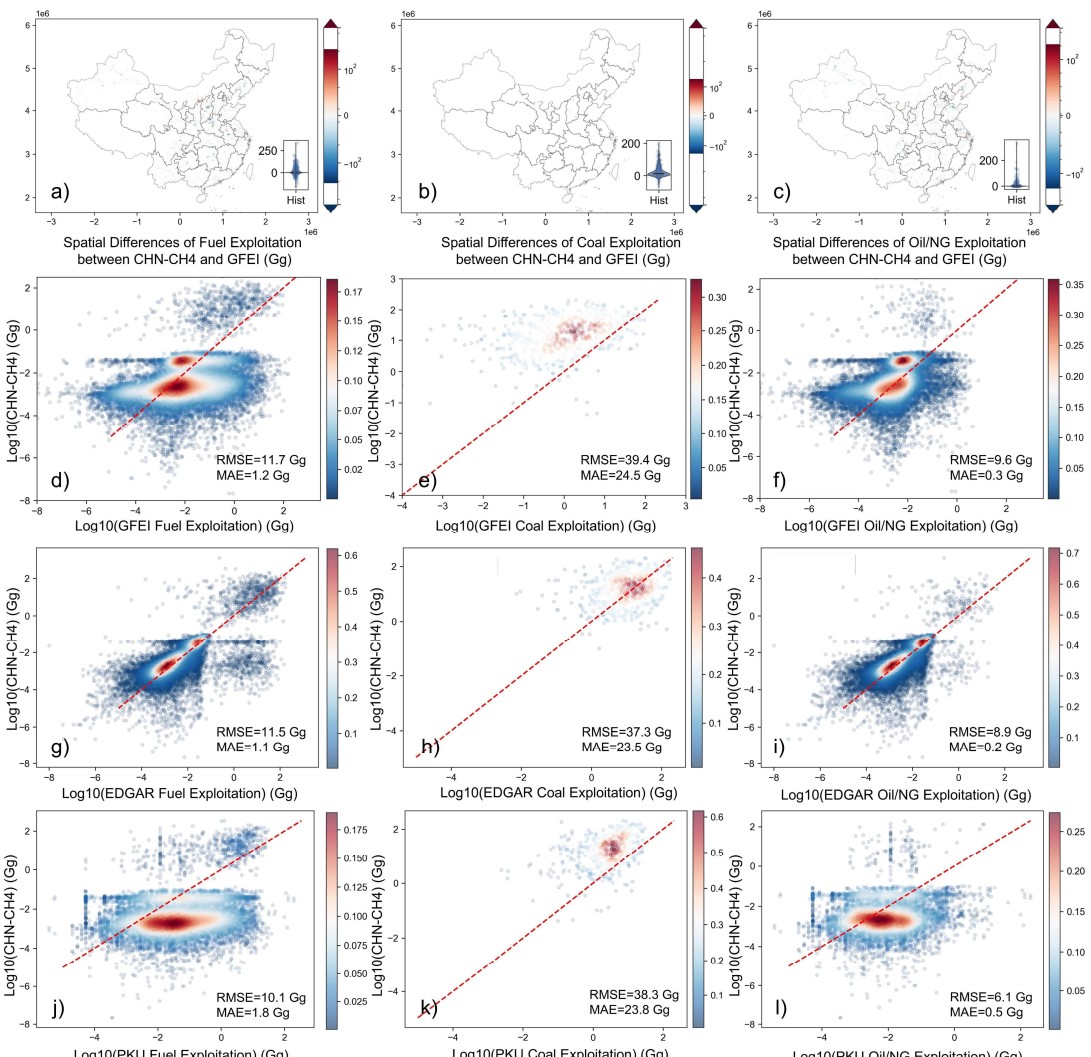

**Figure 4. Pixel-level comparisons between CHN-CH4 and three inventories (GFEI, EDGAR, and PKU-CH4) in the energy sectors for 2019. a-c) show spatial differences between CHN-CH4 and GFEI. d-i) present log-log scatterplots of pixel-level emissions: CHN-CH4 versus GFEI (d–f), EDGAR (g–i), and PKU-CH4 (j-l), respectively.**

The emissions from 26 bottom-up inventory estimates and 14 top-down atmospheric inversion models were selected for the sectoral and national comparisons (Fig. 5). The CHN-CH4 dataset reveals a clear increasing trend in total anthropogenic methane emissions from 1990 to 2020, though with moderate agreement to reference data ($R^2 = 0.11$, $RMSE = 10.5$ Tg, $MAE = 7.6$ Tg) (Fig. 5a). This discrepancy is largely attributable to EDGAR's systematic overestimation, which exceeds CHN-CH4 estimates by at least 36% throughout the study period - consistent with the 30-40% overestimation previously reported by Peng et al. (2016). When EDGAR is excluded, the upscaled national emissions show substantially better agreement ($R^2 = 0.57$, $RMSE = 6.7$ Tg, $MAE = 4.7$ Tg). The overestimation is particularly pronounced in the rice cultivation and wastewater sector, potentially due to EDGAR's use of higher emission factors and assumed continuous flooding rates. In other sectors, this phenomenon is not obvious. Notably, most inventories (including CHN-CH4) show a gradual decline post-2014, although EDGAR's

estimates continued increasing with at a slower rate.

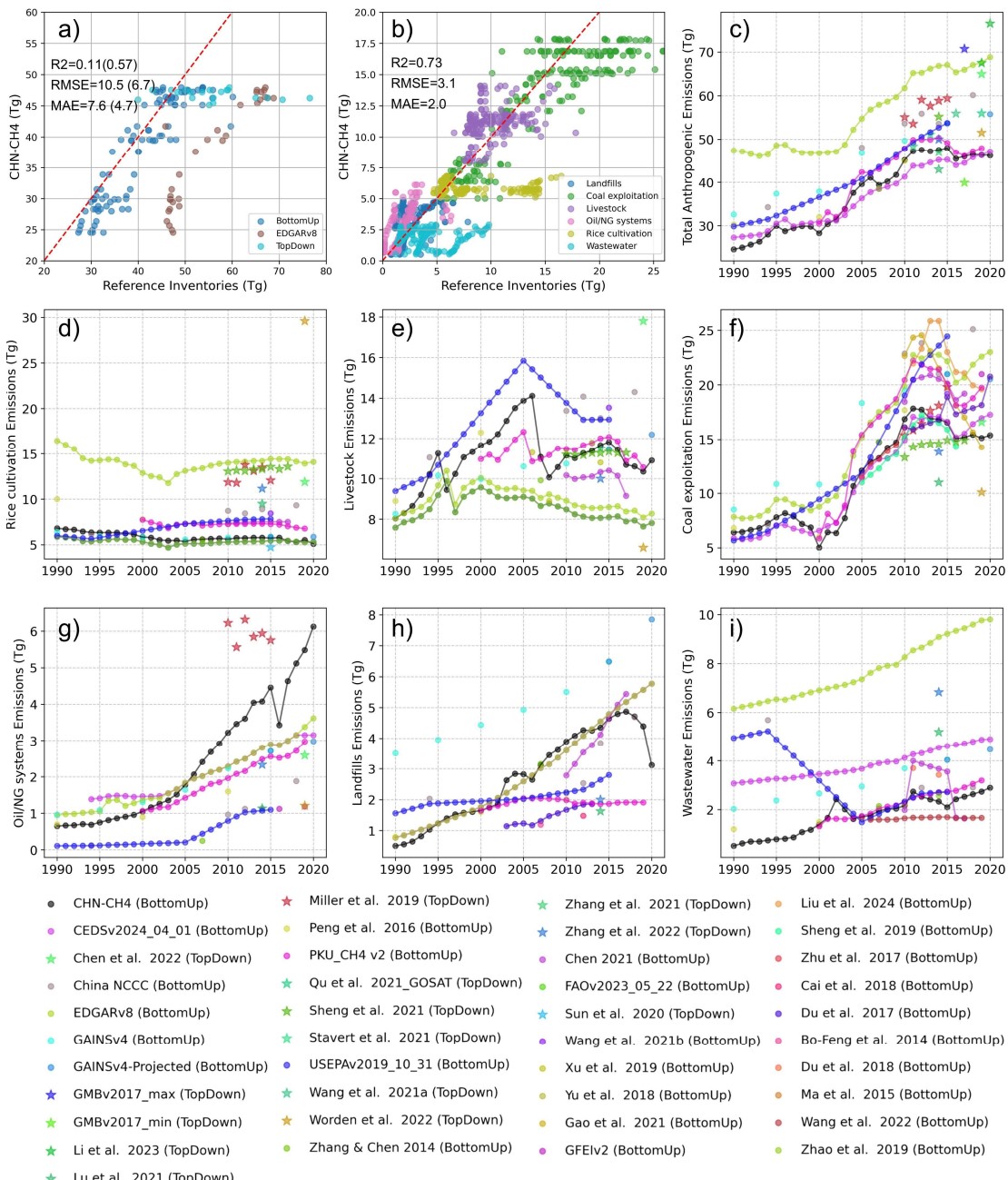

**Figure 5. Sectoral and national comparisons between CHN-CH4 and reference inventories. a) National-level emission comparisons between CHN-CH4 and references, b) Combined sectoral emissions comparison across all inventory sources, c) Variations of total anthropogenic emissions of CHN-CH4 and references, and d-i) Individual sector-specific variations between CHN-CH4 and**

355 **the reference (rice cultivation, livestock, coal exploitation, Oil/NG systems, landfills, and wastewater, respectively). The red line is 1:1 line.**

At the sectoral level, CHN-CH4 exhibits strong consistency with reference inventories, achieving a $R^2$ of 0.73, $RMSE$ of 3.1 Tg, and $MAE$ of 2.0 Tg across all sectors nationwide. These results clearly identify coal mining, livestock, and rice cultivation as China's three largest methane emission sources (Fig. 5b).

CHN-CH4 demonstrates significantly better agreement with bottom-up estimates across both national

and sectoral levels, compared to top-down atmospheric inversions (Fig. 5c-i). The *RMSE* between CHN-CH4 and other bottom-up estimates ranges from 1.3 Tg (landfills, lowest) to 3.7 Tg (rice cultivation, highest), while comparisons with atmospheric inversions show notably higher discrepancies (2.5-8.6 Tg).

**3.2 Spatial patterns and historical emissions of methane emissions**

Figure 6 shows the spatial distribution of total methane emissions and its eight major sectors in China for the year 2020. The total emissions were estimated at 43.4 [31.6–53.4] $Tg$. The largest sources of methane emissions were coal exploitation (14.8 [8.8–20.1] $Tg$), followed by livestock (9.1 [7.6–10.9] $Tg$), and rice cultivation (5.5 [3.9–7.3] $Tg$).

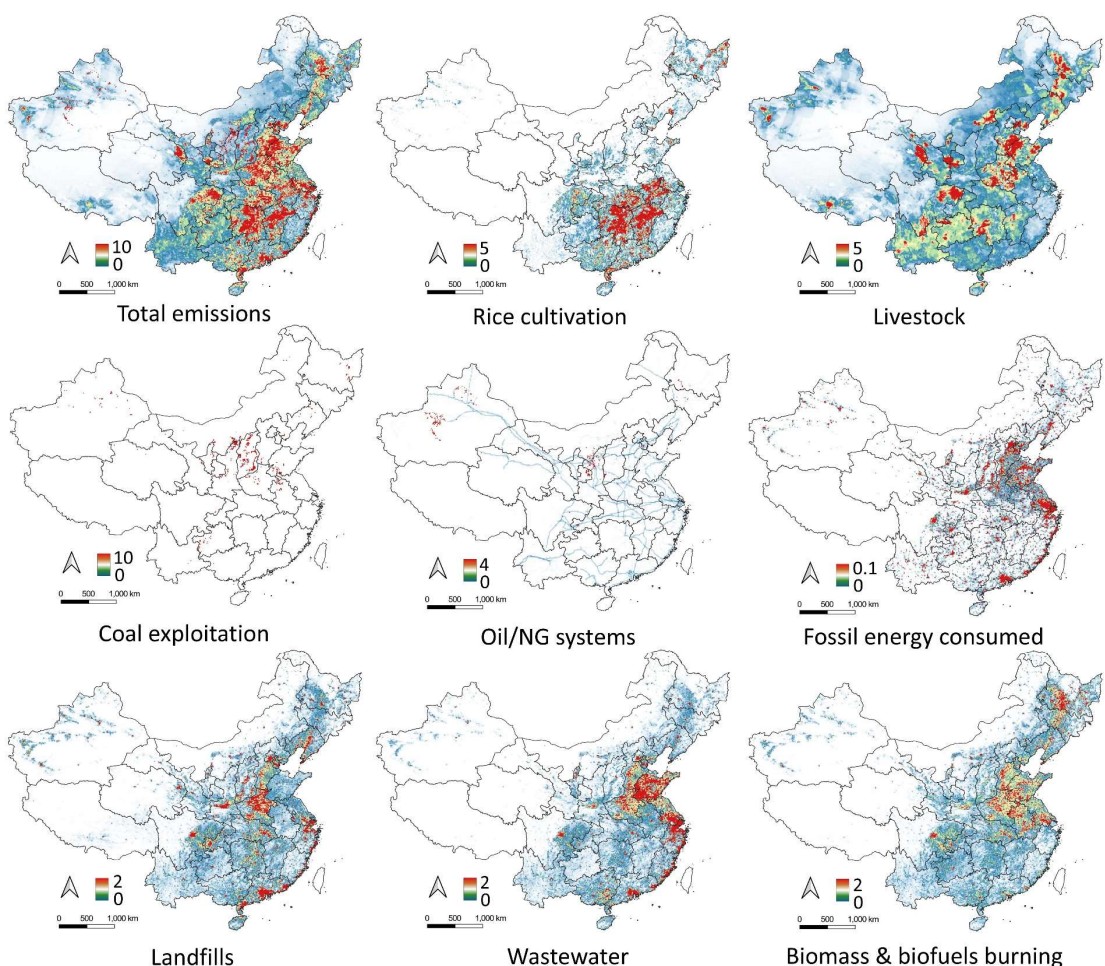

**Figure 6. Spatial distributions of methane emissions from CHN-CH4 at 2020 (unit: $Mg/km^2$).**

The methane emissions decreased from east to west. Provinces in central and northern China, such as Shandong, Henan, Hunan, and Jiangxi, had higher emissions (Fig. 6). In northern China, the spatial distribution of high methane emissions was strongly correlated with areas of fossil energy exploitation, particularly in Shanxi and Xinjiang. In other provinces like Shandong, Henan, Gansu, and Heilongjiang,

emissions were more closely linked to livestock production. Additionally, the emissions from biomass and biofuel burning were mainly located in several yield plateaus, including North China Plain (e.g., Shandong, Henan, and Hebei), Northeast China Plain (e.g., Heilongjiang, Jilin and Liaoning), and Jianghuai Plain (e.g., Jiangsu and Anhui). The rice cultivation contributed to the largest methane

emissions in central China, including Hubei, Hunan and adjacent Jiangxi province. For the eastern and
southern coastal provinces, including Jiangsu, Shanghai, Zhejiang, and Guangdong, methane emissions
were higher due to sectors related to modernization and urbanization, such as wastewater management,
landfills, and fossil fuel consumption. Over the past decades, China has emitted a total of 1156.7 [868.7–
1437.3] $Tg$ of methane. Shanxi, Henan, Guizhou, and Sichuan were the top four emission provinces,
accounting for around 338.8 $Tg$, with 30% of the total emissions.

Table. 5 shows the average methane emissions from various sectors at the national level over three
decades. Total emissions increased from 28.8 [20.3-37.9] $Tg$ to 45.2 [29.1-58.9] $Tg$, with an increase of
over 55%. Agricultural sectors were the largest contributors during the period 1990–2020, but the
proportion of emissions from the energy sector, particularly coal exploitation, rose significantly, making
it the largest source of methane emissions in 2010–2020. In the first decade, emissions from agriculture
sectors contributed about 62% to total methane emissions, but decreased to 42% in the 2010s. In contrast,
emissions from the energy sector grew from 28% to 42%, particularly the coal exploitation, emitting 16.4
[9.8-22.3] $Tg$ methane in 2010-2020. Similar trends were observed in the RIMAP-hist national historical
emissions dataset (Gütschow et al., 2016, 2024). This shift is primarily driven by industrialization,
modernization, and urbanization, which have increasingly relied on fossil fuel consumption. Another
noteworthy trend is the continuous rise in emissions from waste sectors, including landfills and
wastewater, which grew from 6% to 15% over the three decades.

**Table 5. Average methane emissions from 8 major sectors for three decades (units: $Tg$).**

|  | 1990-1999 | 2000-2009 | 2010-2020 |
|---|---|---|---|
| Agriculture | 17.9 [13.5-23.0] | 19.7 [14.8-25.0] | 19.2 [13.2-24.7] |
| Rice cultivation | 6.5 [4.7-8.6] | 5.6 [4.0-7.3] | 5.6 [4.0-7.4] |
| Livestock | 9.7 [8.1-11.7] | 12.2 [10.1-14.7] | 11.2 [9.3-13.4] |
| Biomass & biofuels burning | 1.7 [0.7-2.7] | 1.9 [0.7-3.0] | 2.4 [0.9-3.9] |
| Energy | 8.9 [5.4-12.6] | 12.9 [7.7-18.0] | 19.1 [11.1-26.0] |
| Coal exploitation | 7.9 [4.7-10.8] | 11.4 [6.8-15.5] | 16.4 [9.8-22.3] |
| Oil/NG system | 0.9 [0.7-1.8] | 1.4 [0.9-2.4] | 2.5 [1.3-3.7] |
| Fossil energy consumed | 0.05 [0.02-0.07] | 0.09 [0.04-0.15] | 0.17 [0.07-0.26] |
| Waste | 2.0 [1.4-2.3] | 4.6 [3.2-5.3] | 6.925 [4.8-8.2] |
| Landfills | 1.2 [0.9-1.3] | 2.7 [2.1-3.2] | 4.376 [3.5-5.3] |
| Wastewater | 0.8 [0.6-0.9] | 1.9 [1.1-2.1] | 2.549 [1.3-2.9] |
| Total emissions | 28.8 [20.3-37.9] | 37.2 [25.7-48.3] | 45.2 [29.1-58.9] |

**3.3 Spatiotemporal changes of methane emissions**

Table. 6 shows the temporal changes in methane emissions from major sectors over the three decades
from 1990 to 2020. Overall, methane emissions increased by 18.0 $Tg$ from 1990 to 2020, with the energy
sector, particularly coal exploitation, contributing to about 43% of this increase. Additionally, emissions
from waste sectors, which increased by 5.0 $Tg$ over the 30 years, also played a significant role in the

total rise. The decline in emissions from rice cultivation further highlights the significant impact of economic development on methane emissions. This decrease in rice cultivation emissions was particularly pronounced in southern China (Fig. 7), which benefited from the reform and opening-up policy, leading to higher GDP and more impervious surfaces. Spatially, northern and southern China exhibited contrasting trends. The overall increase in northern China, especially in provinces such as Xinjiang, Shanxi, Henan, Hebei, and Heilongjiang, was primarily driven by emissions from livestock, wastewater, energy exploitation, and biomass and biofuel burning.

**Table 6. Methane emission changes from 8 major sectors for three decades (units: $Tg$).**

|  | 1990-1999 | 2000-2009 | 2010-2020 | 1990-2020 |
|---|---|---|---|---|
| Agriculture | 3.1 | -1.0 | 0.1 | 2.6 |
| Rice cultivation | -0.6 | -0.3 | -0.1 | -1.4 |
| Livestock | 3.5 | -1.1 | -0.3 | 3.0 |
| Biomass & biofuels burning | 0.2 | 0.4 | 0.4 | 1.0 |
| Energy | 1.4 | 11.0 | -1.7 | 10.4 |
| Coal exploitation | 1.2 | 10.2 | -3.0 | 7.9 |
| Oil/NG system | 0.2 | 0.7 | 1.3 | 2.4 |
| Fossil energy consumed | 0.02 | 0.08 | 0.04 | 0.1 |
| Waste | 1.7 | 2.7 | -0.0 | 5.0 |
| Landfills | 1.1 | 2.0 | -0.8 | 2.6 |
| Wastewater | 0.6 | 0.7 | 0. 8 | 2.4 |
| Total emissions | 6.2 | 12.7 | -1.6 | 18.0 |

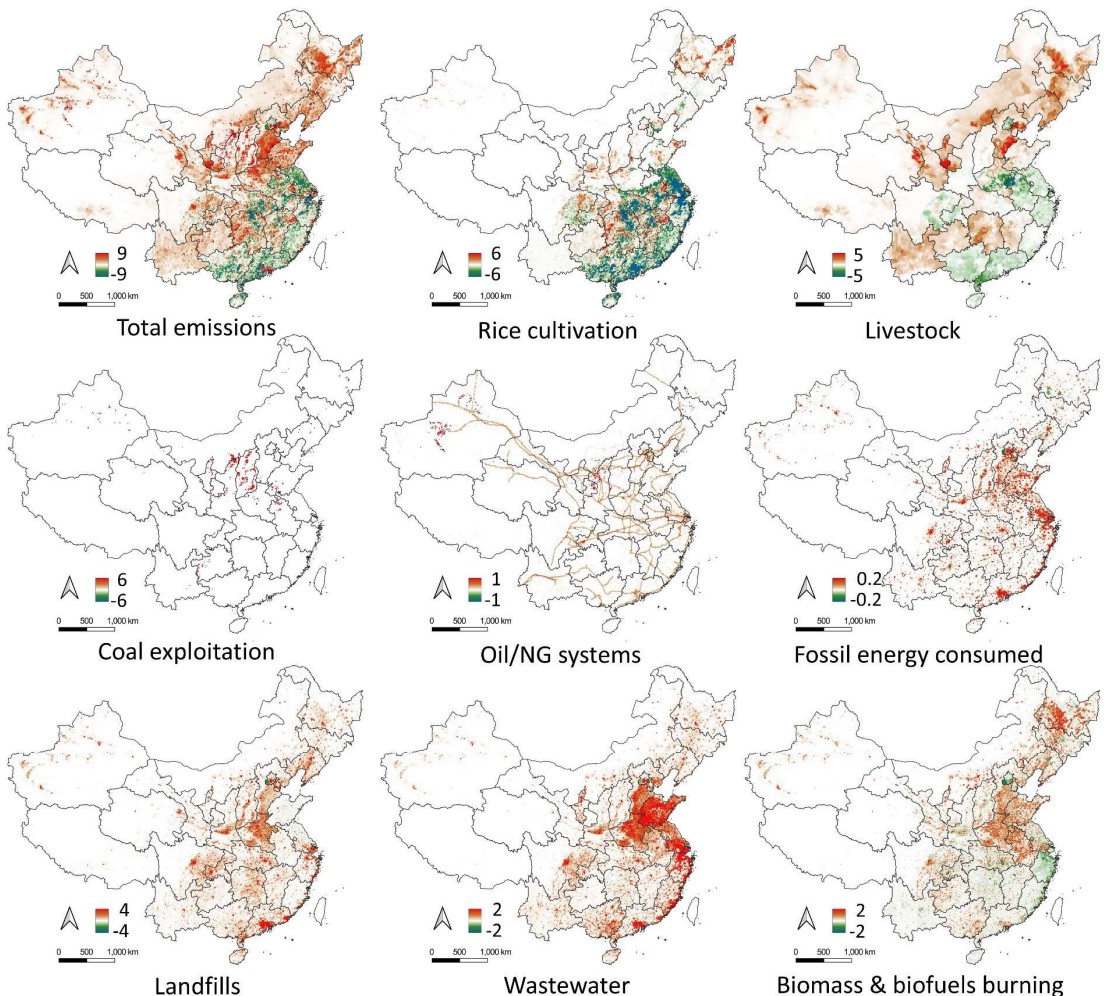

**Figure 7. Spatial changes of methane emissions from 1990 to 2020 (unit: $Mg/km^2$).**

Between 1990 and 2020, the decade from 2000 to 2009 saw the most significant increase in methane emissions, rising by over 12 $Tg$ (Table. 6), largely driven by the rapid economic development. During this period, the energy sectors, particularly coal exploitation, accounted for more than 85% of the increase. In the following decade (2010-2020), under the new government and a series of methane abatement policies, including the Twelfth and Thirteenth Five-Year Plans for Controlling Greenhouse Gas Emissions and the China National Plan for Tackling Climate Change (2014-2020) (NDRC, 2014; State Council of China, 2011; State Council of China, 2016), total methane emissions began to decrease by more than 1.6 $Tg$. Most sectors met their methane reduction targets, with coal exploitation seeing the largest decrease—over 3 $Tg$. This was partly due to the consolidation of China's coal mining industry, where approximately 12,000 outdated mines were closed to improve workplace safety, enhance resource utilization, and reduce overcapacity (State Council of China, 2013; Liu et al., 2024).

**3.4 Policy Implications**

CHN-CH4, as a high-resolution gridded methane emission inventory, provides a critical policy tool for guiding national, subnational, and sector-specific methane mitigation efforts. This dataset can potentially enhance emission tracking accuracy, inform policymakers, and facilitate international cooperation.

CHN-CH4 could be applied to support China's domestic efforts in methane mitigation and climate
change governance. Given China's existing commitments for 2030 carbon emission peaking and 2060
carbon neutrality, even though the primary focus is on carbon, methane, with its high warming potential,
remains a critical yet under-regulated area. The CHN-CH4 inventory provides granular emission data
from 1990 to 2020, can enable the assessment of methane emission and enhance data transparency and
improved reporting under national initiatives including the 14[th] Five Year Plan and the National Methane
Action Plan. On the regional and subnational level, given China's vast regional disparities in methane
emissions (explained in Fig. 6), there is no one-fits-all policy for economy-wide emission reduction.
CHN-CH4 can enable regional-specific policy interventions by identifying emission hotspots and regions
for more targeted mitigation measures, such us identifying high-emission provinces like Shanxi, Guizhou,
Inner Mongolia for coal mining regulations, optimizing reduction strategies in livestock and rice
cultivation in regions, and identifying urban centers with high emission from waste management to
enhance policymaking around landfill and wastewater treatment efficiency. Furthermore, at the city level,
CHN-CH4 can provide evidence for policymaking in urban areas to design more effective methane
abatement programs.

Beyond domestic policy, this dataset can enhance China's role in global methane reduction efforts and
international cooperation. For example, as a pillar in the 2023 US-China Sunnylands Statement, through
enhanced emission tracking and transparency, China and the US can collaborate on effective methane
mitigation. Potential integration between CHN-CH4 and other satellite-based methane monitoring can
improve the MRV capacity and technique globally.

**4 Uncertainty, Limitations and Prospects**

The CHN-CH4 inventory, derived from eight major sectors, still exhibits relatively high uncertainty. This
uncertainty arises from the use of activity data, emission factors, and correction factors. Key sources of
uncertainty include:

- Data Sources and Consistency: The estimates rely on diverse data sources, including remote sensing
  products, statistical yearbooks, and existing datasets. Each of these sources comes with its own level
  of accuracy and consistency, which can vary significantly. This variability poses challenges for
  ensuring the comparability, completeness, and reliability of the CHN-CH4 inventory. Discrepancies
  in data collection methods, temporal and spatial resolution, and reporting standards across these
  sources can introduce uncertainties and potential biases in the emission estimates. Therefore, careful
  consideration and cross-validation of these data sources are essential to enhance the robustness and
  credibility of the CHN-CH4 inventory.
- Data Filling Methods for Spatial and Temporal Gaps: The interpolation methods used to fill data
  gaps introduce additional limitations. Spatially, for instance, the activity data for landfills and
  wastewater lack detailed spatial proxies. Instead of using specific locational data, the distribution of
  GDP and population is employed for regression and prediction purposes, which may not accurately
  reflect the true spatial distribution of emissions. This reliance on indirect proxies can introduce
  significant uncertainties in the spatial allocation of emissions. For instance, our estimate of methane

emissions from landfills in China for 2012 (4.2 Mt) differs from the 1.8 Mt reported by Cai et al. (2018), who estimated the emissions using the activity data from Chinese Ministry of Environmental Protection, provincial environmental protection bureaus and field investigations. This substantial difference suggests that conventional bottom-up approaches can systematically misestimate emissions compared to observational constraints. Temporally, addressing missing data can be even more challenging, as exemplified by livestock data, where global livestock density distributions from selected years and province-level data from various statistical yearbooks were used to fill gaps. This approach, while necessary, may not fully capture the dynamic changes in livestock populations over time and across different regions. It can lead to potential inaccuracies in the temporal trends of methane emissions.

- Lack of Seasonality Consideration: CHN-CH4 currently focuses only on annual changes, which means it does not capture short-term variations that occur on a seasonal basis. This limitation can be significant, as methane emissions can fluctuate considerably throughout the year due to factors such as agricultural practices, temperature changes, and seasonal variations in industrial activities. For example, emissions from rice paddies, livestock, and waste management can vary seasonally, influenced by planting and harvesting cycles, livestock breeding patterns, and seasonal waste generation and treatment processes. Addressing this limitation would be valuable for future research.

- Comprehensibility of methane emissions sectors: CHN-CH4 does not take the methane emissions from abandoned coal mines into consideration. Understanding its emissions and trends is critical for a low-carbon planet with more outdated mines closure. Current methodology still tends to use the default emissions factors, or the ratio of flooded or dry coal mines regionally/globally, which might bring large uncertainty. This sector warrants greater attention, particularly in developing spatially explicit mine status data and dynamic emission factors, given its substantial estimated emissions of 20.1 Tg annually from 2010–2019 (Gao et al., 2021). Existing literature also highlights the underestimation of AMM emissions in China, particularly as current bottom-up estimates often fail to account for their increasing trends (Liu et al., 2024). Additionally, the emissions from oil/NG systems were not further subdivided into upstream, midstream, and downstream processes, due to limited spatial information. However, more detailed estimates for this sector are important to identify key emission sources to targeted mitigation technologies, and enable tailored solutions rather than one-size-fits-all policies.

In the future, integrating high-resolution, high-accuracy remote sensing images with big data analysis offers promising solutions to address these limitations through providing detailed and accurate data for methane emission sources. Over the past five years, significant advancements in remote sensing products related to methane emissions have been achieved. Notable examples include the development of CCD-Rice for rice cultivation monitoring (Shen et al., 2025) and GLW for livestock detection (Gilbert et al., 2018; Ocholla et al., 2024). Additionally, Kolluru et al. (2023) developed a high-resolution (1 km) gridded livestock density database for horses and small ruminants (sheep and goats) in Kazakhstan for 2000–2019, using vegetation proxies, climatic, socioeconomic, topographic, and proximity variables through random forest regression modelling. This study demonstrated the potential of remote sensing technologies for mapping spatial distributions of activity data.

Another critical advantage of remote sensing is its ability to detect methane emission sources that are unregistered in administrative records or omitted from official statistics. This includes sources include closed coal mines, small-scale agricultural activities and other diffuse or poorly documented emission points. Remote sensing technologies, with their extensive spatial coverage and high resolution, can identify and quantify emissions from these overlooked sources, thereby providing a more comprehensive and accurate inventory. Additionally, companies like Google, Baidu, and Bing offer API services that can extract geocoded locations for point-source methane emissions, such as coal mines (Sheng et al., 2019). Utilizing these tools can significantly enhance methane emission inventories by filling data gaps and improving spatial accuracy.

**5 Data and codes availability**

The distribution maps of methane emissions in China (CHN-CH4) and its major eight sectors from 1990 to 2020 are publicly available at https://doi.org/10.5281/zenodo.15107383 (Guo et al., 2025). This website also published the reference inventories for comparison, coming from Behrendt et al. (2025), Bo-Feng et al. (2014), Cai et al. (2018), Chen, (2021), Chen et al. (2022), Du et al. (2017), Du et al. (2018), European Commission and Joint Research Centre (2023), Gao et al. (2021), Hoesly and Smith (2024), Höglund-Isaksson et al. (2020), Li et al. (2023), Liu et al. (2024), Lu et al. (2021), Ma et al. (2015), Miller et al. (2019), Peng et al. (2016), PKU-CH4_v2 (2021), Qu et al. (2021), Scarpelli et al. (2022), Sheng et al. (2019), Sheng et al. (2021), Stavert et al. (2022), Sun et al. (2020), U.S. Environmental Protection Agency (2019), Wang et al. (2021b), Wang et al. (2021a), Wang et al. (2022), Worden et al. (2022), Xu et al. (2019), Yu et al. (2018), Zhang and Chen, 2014), Zhang et al. (2021), Zhang et al. (2022), Zhao et al. (2019), and Zhu et al. (2017). The product is provided in GeoTIFF format with a spatial reference system of Krasovsky_1940_Albers.

**6 Conclusions**

This study introduces CHN-CH4, a spatially aggregated $0.1° \times 0.1°$ anthropogenic methane emission inventory for mainland China annually, covering the period from 1990 to 2020. The inventory integrates long-term remote sensing data, multiple national statistical yearbooks, and methane emission datasets from international organizations. Emissions were estimated using recent regional and localized emission factors, along with IPCC default factors, across eight major sectors, covering the fields of agriculture, energy and waste. CHN-CH4 demonstrates strong consistency with reference inventories—top-down atmospheric observations and bottom-up estimates—at pixel, sectoral, regional, and national scales. This inventory enhances robust comparisons between estimated emissions and observed atmospheric data, improving the accuracy of inverse modelling. Furthermore, CHN-CH4 can complement global climate models such as the Global Change Assessment Model (GCAM) and the Community Earth System Model (CESM) to predict methane emission trends, assess potential climate impacts, and evaluate socioeconomic outcomes under different development and policy scenarios. Moving forward, CHN-CH4 can be further integrated into national reporting frameworks, climate modeling efforts, and even carbon

market mechanisms to enhance data transparency and methane mitigation efforts. Additionally, it aims to increase public awareness of environmental and health implications and support effective tracking and policy-making for methane emission mitigation.

## Author contributions

FD and YZ conceptualized the study. FG developed and implemented the computer code, performed the analysis, validation, and visualized the results, and wrote the manuscript. FG, FD, PG, and YZ edited and revised the manuscript.

## Competing interests

The authors declare that they have no conflict of interest.

## Disclaimer

Publisher's note: Copernicus Publications remains neutral with regard to jurisdictional claims made in the text, published maps, institutional affiliations, or any other geographical representation in this paper.

While Copernicus Publications makes every effort to include appropriate place names, the final responsibility lies with the authors.

## Financial support

This research was supported by "Global Methane Hub" project and Research Grants Council-Strategic Topics Grant STG2/P-705/24-R.

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
