# Peer review of "CHN-CH4: A Gridded (0.1°× 0.1°) Anthropogenic Methane Emission Inventory of China from 1990 to 2020"

_Earth System Science Data, 2025_

## Author Response (AR1)

**Response to Reviewer 1**

Our comments are inset in blue colour following each point of the reviewers. The text quoted directly from the revised manuscript is set in italics. The line numbers cited in our response refer to the revised manuscript with no changes marked.

The authors create a high resolution bottom-up inventory of methane emissions in China, offering a valuable dataset for the community. My comments are concerned primarily with (1) clarifying methodology and (2) strengthening evaluation against other datasets. I support publication after these comments are addressed.

Response: We sincerely appreciate the Reviewer's valuable comments and suggestions, which help us significantly improve the manuscript. In the revised manuscript, we addressed each of these concerns: 1) enhancing the methodological descriptions with particular attention to handling missing data; 2) conducting a comprehensive comparisons of CHN-CH4 with EDGAR (v8), PKU-CH4 (v2) and GFEI spatially; 3) performing an extended evaluation using 26 bottom-up estimates and 14 top-down atmospheric inversions to validate CHN-CH4 at both sectoral and national levels for 1990-2020; 4) thoroughly discussing uncertainty sources while refining all figures and clarifying text throughout.

Major comments:

1. Figure 3: It is not clear to me what is being compared in the scatter plots d-i. Are these all individual grid cells in China for a given year (2000/2019)? For 2000-2009, is this a decadal average or all data for each year? It is also clear that there is a density plot, but there is no colorbar showing how many points are combined at the yellow saturation. Perhaps this would be better shown on a log-log plot so lower emissions grid cells are more visible and easier to compare.

Response: In the initial submission, the data for 2000–2009 represents annual average across all data, and we displayed only a randomly selected subset of 10,000 points for clarity because of the low emission values in many grid cells. As suggested, we improved the figure and adopted a log-log plot to better illustrate the differences between our results and those from EDGAR, PKU-CH4, and GFEI (please see Figure 3 and Figure 4 in the revised manuscript).

2. Lines 307-311: How do you know EDGAR is overestimating these emissions? Bottom-up

estimation is complicated and can have errors, and top-down quantification similarly has errors; our uncertainties are generally quite large. In particular, I disagree with line 311; I think this paper would greatly benefit from a more detailed comparison with EDGAR at sector, regional, and national levels. Disagreement between your estimates is expected, but it is important to understand why your inventory disagrees.

Response: We acknowledge that bottom-up estimates inherently carry high uncertainties, and thus it would be oversimplistic to categorically assert an overestimation in EDGAR. In the revised manuscript, we provided a more comprehensive comparison between CHN-CH4 and EDGAR across pixel, sectoral, regional, and national levels. Spatially, CHN-CH4 exhibits strong agreement with EDGAR, with most grid cells aligning closely along the 1:1 line (Figure 3 below). A detailed histogram analysis further reveals that spatial differences are predominantly centred around zero. However, notable deviations still exist. CHN-CH4 exhibits higher emissions in North China (e.g., Shandong and Henan) but lower estimates in energy-intensive provinces (e.g., Shanxi and Sichuan), major rice-growing regions (e.g., Hunan and Jiangxi), and developed coastal areas. At the sectoral and national levels, we recollected 40 existing inventories (26 bottom-up estimates and 14 top-down inversions) for comparison. CHN-CH4 provides continuous coverage from 1990 to 2020 at a higher spatial resolution than most references. In terms of total anthropogenic emissions, EDGAR exceeds CHN-CH4 by over 36% and PKU-CH4 by over 30%, primarily due to overestimations in rice cultivation and wastewater sectors. For other sectors, EDGAR's magnitude and variability align well with the reference inventories. Given these findings, we refrained from conclusively attributing overestimation to EDGAR and revised our manuscript accordingly. The updated text now reads (Lines 309-315):

*Spatial validation against EDGAR v8 and PKU-CH4 v2 datasets for the years 2000, 2009, and 2019 demonstrates strong agreement with CHN-CH4 along the 1:1 line (Fig. 3). The higher $R^2$ between CHN-CH4 and EDGAR indicate greater spatial consistency compared to PKU-CH4. Notable regional differences persist, with CHN-CH4 showing lower emission estimates than both EDGAR and PKU-CH4 in energy-intensive provinces (e.g., Shanxi and Sichuan) and major rice-growing regions (e.g., Hunan and Jiangxi). Particularly, the deviations from EDGAR are more pronounced than those from PKU-CH4.*

[Figure]

*Figure 3. Pixel-level comparisons between CHN-CH4 and EDGAR/PKU-CH4. a-f) represent the comparison between CHN-CH4 and EDGAR v8, while g-l) represent the comparison between CHN-CH4 and PKU-CH4 v2 at year 2000, 2009, and 2019. 'Hist' in each spatial map represents the histogram of the differences between CHN-CH4 and EDGAR/PKU-CH4, with the unit Gg. The bottom-right subfigure in each log-log plot presents threshold-dependent performance metrics, demonstrating how RMSE and $R^2$ vary when excluding grid cells over specific emission thresholds.*

Lines 338-349 read as:

*26 bottom-up inventory estimates and 14 top-down atmospheric inversion models were selected for the sectoral and national comparisons (Fig. 5). The CHN-CH4 dataset reveals a clear increasing trend in total anthropogenic methane emissions from 1990 to 2020, though with moderate agreement to reference data ($R^2$ = 0.11, RMSE = 10.5 Tg, MAE = 7.6 Tg) (Fig. 5a). This discrepancy is largely attributable to EDGAR's systematic overestimation, which exceeds CHN-CH4 estimates by at least 36% throughout the study period - consistent with the 30-40% overestimation previously reported by Peng et al. (2016). When EDGAR is excluded, the upscaled national emissions show substantially better agreement ($R^2$ = 0.57, RMSE = 6.7 Tg, MAE = 4.7 Tg). The overestimation is particularly pronounced in the rice cultivation and wastewater sector, potentially due to EDGAR's use of higher emission factors and assumed continuous flooding rates. In other sectors, this phenomenon is not obvious. Notably, most inventories (including CHN-CH4) show a gradual decline post-2014, although EDGAR's estimates continued increasing with at a slower rate.*

[Figure]

*Figure 5. Sectoral and national comparisons between CHN-CH4 and reference inventories. a)*
*National-level emission comparisons between CHN-CH4 and references, b) Combined sectoral*
*emissions comparison across all inventory sources, c) Variations of total anthropogenic*
*emissions of CHN-CH4 and references, and d-i) Individual sector-specific variations between*
*CHN-CH4 and the reference (rice cultivation, livestock, coal exploitation, Oil/NG systems,*

*landfills, and wastewater, respectively). The red line is 1:1 line.*

3. In addition to your scatterplots in Figure 3, I'd also like to see a spatial comparison with other inventories, especially EDGAR and also perhaps some leading sectoral inventories like GFEI for fossil fuels (especially coal). It would also be interesting to see spatial comparisons with a top-down posterior estimates. Do your maps generally agree on the spatial patterns of emissions, or are there significant differences? This could perhaps be shown in a difference plot like Figure 5.

Response: In the revised manuscript, we enhanced the spatial analysis with three bottom-up inventories (GFEI, EDGAR, and PKU-CH4). Grid-cell-level comparisons with top-down posterior estimates were not included due to data accessibility constraints. Instead, we conducted comprehensive sectoral and national-level comparisons using published top-down estimates from the literature (see Figure 5). This multi-scale evaluation strengthens the robustness of our CHN-CH4 emission estimates. For the three bottom-up inventories, we made comparison across three energy sectors: (1) total fuel exploitation, (2) coal exploitation, and (3) oil/natural gas (NG) exploitation (see Figure 4 below). Results demonstrate well spatial consistency between CHN-CH4 and GFEI for total fuel exploitation (with differences clustered around zero), while the main discrepancies originate from coal mining (RMSE: 39 Gg; MAE: 24.5 Gg). In contrast, the oil/NG sector demonstrates much closer alignment, as evidenced by the log-log plot. Among the three inventories, EDGAR exhibits the best spatial distribution agreement with CHN-CH4 (with data points densely concentrated along the 1:1 line), whereas PKU-CH4 provides closer emission magnitude estimates (lower RMSE and MAE values). We added these descriptions in the revised manuscript. The updated text now reads (Lines 324-334):

*Further comparison with the Global Fuel Exploitation Inventory (GFEI) across energy sectors shows well spatial agreement for oil/NG systems but larger discrepancies in coal mining (RMSE > 39.4 Gg, MAE >24.5 Gg) (Fig. 4). In contrast, the oil/NG sector demonstrates much closer alignment, as evidenced by the log-log plot (Fig. 4c and 4f). Compared to GFEI and PKU-CH4, CHN-CH4 exhibits stronger spatial consistency with EDGAR, as evidenced by a higher concentration of points along the 1:1 line. The spatial discrepancies between GFEI and EDGAR primarily arise from differences in their underlying data sources for fuel exploitation; EDGAR's coal and oil/gas system locations were adopted in this study. Despite these variations, the overall alignment supports the reliability of CHN-CH4 in identifying key sectoral and regional emission*

*patterns.*

[Figure]

***Figure 4. Pixel-level comparisons between CHN-CH4 and three inventories (GFEI, EDGAR, and PKU-CH4) in the energy sectors for 2019. a-c) show spatial differences between CHN-CH4 and GFEI. d-i) present log-log scatterplots of pixel-level emissions: CHN-CH4 versus GFEI (d–f), EDGAR (g–i), and PKU-CH4 (j-l), respectively.***

4. On that note, what version of EDGAR do you use for your comparison? Table 1 suggests V8 but it isn't clear from the citations. How do your comparisons look with different versions of EDGAR? V8 for example tends to have more emissions from point sources.

Response: We compared the latest version of EDGAR (v8), which incorporates more detailed point-source emissions, particularly for industrial processes related to fossil fuel use. Due to limited spatial activity data for industrial emissions in China, we allocated the fossil fuel consumption data using economic proxies such as GDP and population density, rather than distinguishing specific industrial processes (e.g., chemical or metal production). For spatial comparison with CHN-CH4, we followed the IPCC 2006 guidelines and aggregated EDGAR's anthropogenic emissions to ensure consistency in categorization, including agriculture sectors (*AGS - Agricultural soils, AWB - Agricultural waste burning, ENF - Enteric fermentation, and MNM - Manure management),* energy sectors (*PRO_COAL - Fuel exploitation COAL, PRO_GAS - Fuel exploitation GAS, PRO_OIL - Fuel exploitation OIL, REF_TRF - Oil refineries and Transformation industry, TRO - Road transportation, TNR_Other - Railways, pipelines, off-road transport*), and waste sectors (*WWT - Waste water handling*, and *SWD_LDF - Solid waste landfills)*. Since our analysis did not include emissions from solid waste incineration, this sector was excluded from the waste-related comparisons. In the revised manuscript, we have expanded the details of our comparison with EDGAR to enhance clarity and minimize potential ambiguities.

5. I would like more information and discussion of the extrapolation and interpolation methodologies used in this paper. It seems like the authors are dependent on simple linear regression for e.g. landfills (255-261), regressing against GDP to predict landfills before yearbook data begin in 2003. I see that on a national basis the correlation in Figure S2 is strong, but is this also true at the provincial level? Similarly, could the authors clarify how missing data are handled, across all sectors? Are the methods the authors use to handle these data in line with previous work?

Response: We have expanded the methodology and discussion sections to provide better clarity regarding our approach to landfills sector. Since the gridded activity data is unavailable, we adopted the methodology of Peng et al. (2016) to allocate provincial statistical data to gridded cells using linear relationships. However, since provincial landfill data were only available from 2003 onward, we employed the following approach to address missing data for earlier years (1990-2002). First, we estimated national landfill amounts by establishing the linear relationship with GDP. As shown in the figure below, the national trend (black line) falls within the 95% confidence interval of provincial relationships (red line), demonstrating that the national pattern effectively captures provincial variations. In this case, we applied the national GDP-landfill relationship to estimate

1990-2002 national totals, and then allocated these estimates to provinces using their 2003-2008 average contribution to national landfill amounts. Later, the provincial landfills for the whole period (1990-2020) were distributed to grid cells using the spatial distribution of GDP, and then the corresponding methane emissions were calculated from the gridded data.

[Figure]

The methods to handle missing data in landfills were in line with previous work (see Figure 2 in the manuscript). Our analysis began by establishing temporal anchors at the earliest (t1) and most recent years (t2) with available statistical data for each province and sector. We employed linear interpolation to estimate missing values between these anchor years based on historical trends. For periods preceding t1 and following t2, national-level statistics were prioritized, and then provincial data were allocated according to their average proportion to the national total over the nearest five-year period. This five-year period was chosen to align with China's Five-Year Plan cycles, given the relative stability of economic development policies. This approach provides a reasonable estimation method when detailed historical provincial data are unavailable, while maintaining alignment with national totals and provincial development patterns.

Minor comments:

6. Could you describe the methodology in the left portion of Figure 1 in more detail (at bottom

left)? I don't understand what t1 and t2 are, for example.

Response: t1 and t2 represents the earliest and most recent available year that we can get the data from statistical yearbook. In the revised manuscript, we added more information to describe the figure (please see Lines 79-83).

7. Abstract (and elsewhere, e.g. line 358 and Table 5): Do you really have 3 decimal points of confidence in your estimates? I recommend rounding.

Response: We agree. As most literature have done, we have adopted one decimal place for reporting confidence values throughout the manuscript. We have carefully reviewed the entire document to ensure consistency in this reporting format and to eliminate any potential ambiguities in our presentation of results.

8. From your dataset, I see that your data is at annual temporal resolution, but this is not clear in the text of the paper. Could you clarify this?

Response: We modified the sentence in the introduction section to clarify the temporal resolution of CHN-CH4 (please see Lines 61-64).

9. Throughout, e.g. lines 182-183, 224, 310: Are the years 1980-2010 correct here or do you mean 1990-2020? If 1980-2010 is correct, how do you handle 2010-2020?

Response: We have corrected the time period from our initial version to the accurate range of 1990–2020 throughout the entire manuscript, including all relevant figures, tables, and methodological descriptions.

10. Line 234: EF for coal combustion is missing

Response: We added the EF for coal combustion, $1 \; kgCH_4TJ^{-1}$ as suggested by IPCC (2006).

11. Line 260-261: The authors grid landfill data using GDP. I am not familiar with landfills in China but I wonder if this a reasonable assumption? Perhaps the authors could compare with estimates that incorporate more spatial information, like this one: Bofeng Cai et al. CH4 mitigation potentials from China landfills and related environmental co-benefits. Adv.4,eaar8400(2018).DOI:10.1126/sciadv.aar8400

Response: While we acknowledge the inherent uncertainties, our assumption of using GDP as a proxy for landfill CH4 emissions appears reasonable given the well observed correlation between

national/provincial landfill data and GDP, particularly in the absence of precise geolocated landfill data. The Reviewer's reference to prior literature (reporting 1.84 Mt CH4 from landfills in 2012 from Chinese Ministry of Environmental Protection, provincial environmental protection bureaus and field investigations) highlights an important discrepancy with our estimate of 4.2 [3.7-5.1] Mt. This substantial difference suggests that conventional bottom-up approaches may systematically misestimate emissions compared to observational constraints. We have expanded our discussion of this finding in the revised manuscript. The updated text now reads (Lines 464-471):

*Instead of using specific locational data, the distribution of GDP and population is employed for regression and prediction purposes, which may not accurately reflect the true spatial distribution of emissions. This reliance on indirect proxies can introduce significant uncertainties in the spatial allocation of emissions. For instance, our estimate of methane emissions from landfills in China for 2012 (4.2 Mt) differs from the 1.8 Mt reported by Cai et al. (2018), who estimated the emissions using the activity data from Chinese Ministry of Environmental Protection, provincial environmental protection bureaus and field investigations. This substantial difference suggests that conventional bottom-up approaches can systematically misestimate emissions compared to observational constraints.*

12. Figure captions in general should be more descriptive of figure content.

Response: We have carefully revised all figure captions in the manuscript to ensure they provide clearer descriptions and remove ambiguous language to better align with the visual presentation.

**Response to Reviewer 2**

Our comments are inset in blue colour following each point of the reviewers. The text quoted directly from the revised manuscript is set in italics. The line numbers cited in our response refer to the revised manuscript with no changes marked.

Guo et al presented (1990-2020) long-term, high-resolution emission inventory for mainland China. Building long-term methane emission inventory is hard work and the efforts by the authors are quite commendable. I also appreciate that the carbon community will have one more regional inventory to use/evaluate. A unique advantage of this work is the authors used statistical yearbook and remote sensing data to improve the temporal coverage.

Response: We sincerely appreciate the Reviewer's constructive suggestions. In the revised manuscript, we have significantly strengthened the revised manuscript by: 1) enhancing methodological descriptions on the usage of remote sensing products; 2) conducting comprehensive comparisons of CHN-CH4 with EDGAR v8, PKU-CH4 v2, and GFEI inventories; 3) performing extensive validation using 26 bottom-up estimates and 14 top-down inversions at both sectoral and national levels; and 4) thoroughly discussing uncertainty sources while refining all figures and clarifying text throughout.

1. My major concerns are on the spatial distributions. For the spatial distributions, the authors took them from existing inventories for some source sectors (FAO inventory for livestock, EDGAR inventory for coal, oil, gas, to some extent). Therefore, the authors found a better consistency with EDGAR than PKUv2, which is thus as expected. I was wondering if the authors can explicitly show maps between your results and EDGAR, and discuss in detail the extent to which your product has improved in spatial accuracy compared to existing inventories (e.g., EDGAR, as the authors stated in the Introduction, which is part of the motivation of this work). If the authors used spatial distribution from existing inventories, which are known to have spatial bias, the novelty of this work and the accuracy of this product demand more clarifications. I would suggest that the authors elaborate (in both text and figures) on this point. Doing so would improve the clarity and benefit the future readers and users of your product.

Response: In the revised manuscript, we enhanced the spatial analysis by incorporating existing inventories (GFEI, EDGAR, and PKU-CH4) for a comprehensive evaluation of CHN-CH4 (see

Figures 3 and 4 below). Among these, EDGAR shows the strongest spatial distribution agreement with CHN-CH4 (points clustered along the 1:1 line), while PKU-CH4 provides closer emission estimates. Spatially, CHN-CH4 displays higher emissions in North China (e.g., Shandong and Henan) but lower estimates in energy-intensive provinces (e.g., Shanxi and Sichuan), major rice-growing regions (e.g., Hunan and Jiangxi), and developed coastal areas (Figure 3). These discrepancies arise primarily from CHN-CH4's higher livestock emissions and lower estimates for coal mining, rice cultivation, and wastewater sectors. At the sectoral and national levels, we compiled 26 bottom-up estimates and 14 top-down inversions for comparison (Figure 5). The CHN-CH4 dataset reveals a clear increasing trend in total anthropogenic methane emissions from 1990 to 2020, though with moderate agreement to reference datasets. This divergence is largely driven by EDGAR's systematic overestimation, which exceeds CHN-CH4 by over 36% and PKU-CH4 by 30–40% (Peng et al., 2016), mainly due to overestimations for rice cultivation and wastewater. For other sectors, EDGAR's magnitude and variability align well with other inventories. We expanded the discussion of these findings in the revised manuscript and rewrote the Section 3.1, to better contextualize the inventory differences and their implications.

[Figure]

*Figure 3. Pixel-level comparisons between CHN-CH4 and EDGAR/PKU-CH4. a-f) represent the comparison between CHN-CH4 and EDGAR v8, while g-l) represent the comparison between CHN-CH4 and PKU-CH4 v2 at year 2000, 2009, and 2019. 'Hist' in each spatial map represents the histogram of the differences between CHN-CH4 and EDGAR/PKU-CH4, with the unit Gg. The bottom-right subfigure in each log-log plot presents threshold-dependent performance metrics, demonstrating how RMSE and $R^2$ vary when excluding grid cells over specific emission thresholds.*

[Figure]

*Figure 4. Pixel-level comparisons between CHN-CH4 and three inventories (GFEI, EDGAR, and PKU-CH4) in the energy sectors for 2019. a-c) show spatial differences between CHN-CH4 and GFEI. d-i) present log-log scatterplots of pixel-level emissions: CHN-CH4 versus GFEI (d–f), EDGAR (g–i), and PKU-CH4 (j-l), respectively.*

[Figure]

***Figure 5. Sectoral and national comparisons between CHN-CH4 and reference inventories. a)***
***National-level emission comparisons between CHN-CH4 and references, b) Combined sectoral***
***emissions comparison across all inventory sources, c) Variations of total anthropogenic***
***emissions of CHN-CH4 and references, and d-i) Individual sector-specific variations between***

*CHN-CH4 and the reference (rice cultivation, livestock, coal exploitation, Oil/NG systems, landfills, and wastewater, respectively). The red line is 1:1 line.*

2. Another concern is the remote sensing dataset. The authors highlighted in the abstract and Fig. 1 that satellite remote sensing are substantially used in their work. But I failed to find such use in a clear way. For example, for rice paddies, the authors claimed that 'Due to the limitations of existing satellite products, which do not cover the entire period from 1990 to 2020, we used two datasets for gridded rice cultivation areas annually: CCD-Rice for the period 1990-2016 (Shen et al., 2024) and ChinaCP for the 115 period 2017-2020 (Qiu et al., 2022).' Therefore, satellite data is not used at least in rice paddy identification. Please clarify in detail how satellite remote sensing is used for the source sectors.

Response: In the revised manuscript, we added detailed descriptions of the two gridded rice paddy datasets used in our analysis. The CCD-Rice dataset (Shen et al., 2024) was derived from Landsat Collection 2 Level-2 Science Products at 30 m spatial resolution, utilizing shortwave infrared bands (B5 of Landsat 5/7 and B6 of Landsat 8). This dataset demonstrates high accuracy, with provincial-level distribution maps showing an average overall accuracy of 89.61% and strong coefficients of determination ($R^2 = 0.85$ for single-season rice and 0.78 for double-season rice) when validated against ground samples. The ChinaCP dataset (Qiu et al., 2022) was developed from MODIS imagery at 500 m resolution using phenology-based mapping algorithms. Validation against ground truth data revealed an overall accuracy of 89%, with excellent agreement to statistical data ($R^2 \geq 0.89$). We have incorporated this information in the revised manuscript (Lines 118-127) to provide readers with clear documentation of our data sources, their spatial resolutions, and validation metrics, ensuring full transparency regarding the foundational datasets used in our analysis. The updated text now reads (Lines 116-126):

*Due to the limitations of existing satellite products, which do not cover the entire period from 1990 to 2020, we used two datasets for gridded rice cultivation areas annually: CCD-Rice for the period 1990-2016 (Shen et al., 2024) and ChinaCP for the period 2017-2020 (Qiu et al., 2022). The CCD-Rice dataset was derived from Landsat Collection 2 Level-2 Science Products at 30 m spatial resolution, with provincial-level distribution maps showing an average overall accuracy of 89.61% and strong coefficients of determination ($R^2 = 0.85$ for single-season rice and 0.78 for double-season rice). The ChinaCP dataset was developed from MODIS imagery at 500 m resolution using*

*phenology-based mapping algorithms. The validation against ground truth data revealed an overall accuracy of 89%, with excellent agreement to statistical data (R² ⩾ 0.89). The accuracy is further applied to evaluate the uncertainty caused by rice paddy area. These datasets were then resampled into 0.1° by 0.1° gridded maps*

Reference:

Shen, R., Peng, Q., Li, X., Chen, X., & Yuan, W. (2024). CCD-Rice: A long-term paddy rice distribution dataset in China at 30 m resolution. Earth System Science Data Discussions, 2024, 1-33. https://doi.org/10.5194/essd-2024-147

Qiu, B., Hu, X., Chen, C., Tang, Z., Yang, P., Zhu, X., ... & Jian, Z. (2022). Maps of cropping patterns in China during 2015–2021. Scientific data, 9(1), 479. https://doi.org/10.1038/s41597-022-01589-8

Minor comments.

3. Line 14: accumulative methane emissions are not very meaningful here. I suggest using the annual average instead.

Response: We revised this as suggested. Please see Line 14.

4. Line 29: livestock is part of agricultural activities. Re-phrase it here.

Response: In the updated version, we modified this sentence. Please see Line 29.

5. Sect. 2.2.1: I was curious if the authors included abandoned coal mines, as Qiang Liu et al., (2024), https://www.nature.com/articles/s41558-024-02004-3, highlighted the big role of it.

Response: The methane emissions from abandoned coal mines (AMM) are not included in our inventory, due to limited data availability on key parameters (e.g., residual gas ratios, site-specific decay rates, and geological conditions). We acknowledge the importance of distinguishing coal mine types (e.g., abandoned vs. active) in methane emission inventories. Recent studies, such as the Global Methane Tracker 2024 suggest that abandoned mines alone may contribute over 4.7 Mt CH4/year in China, underscoring their significance in national budgets. Existing literature also highlights the underestimation of AMM emissions in China, particularly as current bottom-up estimates often fail to account for their increasing trends. In the revised manuscript, we expanded the discussions in Section 4, where we emphasize that future work should prioritize: 1) developing

spatially resolved datasets of mine status and closure dates; 2) incorporating dynamic emission factors for abandoned mines; and 3) integrating these sources into gridded inventories We agree that this represents a critical gap requiring attention in subsequent inventory versions. Now Lines 484-491 read:

*CHN-CH4 does not take the methane emissions from abandoned coal mines into consideration. Understanding its emissions and trends is critical for a low-carbon planet with more outdated mines closure. Current methodology still tends to use the default emissions factors, or the ratio of flooded or dry coal mines regionally/globally, which might bring large uncertainty. This sector warrants greater attention, particularly in developing spatially explicit mine status data and dynamic emission factors, given its substantial estimated emissions of 20.1 Tg annually from 2010– 2019 (Gao et al., 2021). Existing literature also highlights the underestimation of AMM emissions in China, particularly as current bottom-up estimates often fail to account for their increasing trends (Liu et al., 2024).*

6. Sect. 2.2.2: Can the authors elaborate on how you assign emission to midstream and downstream emissions? I believe it's missing from this section right now.

Response: We did not further subdivide this sector into upstream, midstream, and downstream processes, in the absence of spatial data on midstream and downstream emissions from oil and NG systems. Instead, we treated them as an aggregated source under IPCC subcategory 1B2 (Fugitive Emissions from Oil and Gas), adopting the methodologies from Schwietzke et al. (2014) and Peng et al. (2016). Our approach applied average emission factors for fugitive methane from China's oil and natural gas systems, encompassing emissions from venting, flaring, exploration, production, upgrading, transport, refining/processing, transmission, and storage. However, distinguishing emissions across upstream, midstream, and downstream processes is critical for identifying key emission sources, enabling targeted mitigation strategies rather than one-size-fits-all policies. We added one sentence in the Section 2.2.2 to clarify the methodology, and expanded the Section Uncertainties to address this limitation and outline future work to improve spatial allocation in emission inventories.

Reference:

Schwietzke, S., Griffin, W. M., Matthews, H. S., & Bruhwiler, L. M. (2014). Global bottom-up

fossil fuel fugitive methane and ethane emissions inventory for atmospheric modeling. ACS Sustainable Chemistry & Engineering, 2(8), 1992-2001. https://doi.org/10.1021/sc500163h

Peng, S., Piao, S., Bousquet, P., Ciais, P., Li, B., Lin, X., ... & Zhou, F. (2016). Inventory of anthropogenic methane emissions in mainland China from 1980 to 2010. Atmospheric Chemistry and Physics, 16(22), 14545-14562. https://doi.org/10.5194/acp-16-14545-2016

Lines 115: I think we should consider uncertainties from both rice area and emission factors. Currently the authors only considered emission factor uncertainties, which is not comprehensive.

Response: The CCD-Rice and ChinaCP datasets exhibit average overall accuracies of 89.61% and 89%, respectively, based on the validation using ground observations. In the revised manuscript, we incorporated these accuracy values to estimate the uncertainties in rice paddy area mapping. For grid cell $G$ in 2010, $A_{G,i}$ is the rice paddy area for season $i$ (where $i \epsilon \{early, middle \ and \ late\}$). Considering the rice area uncertainty of 89.61% accuracy, the error bounds for each seasonal area can be expressed as:

$$\left[A_{G,i} - A_{G,i} * (1 - 89.61\%), A_{G,i} + A_{G,i} * (1 - 89.61\%)\right]$$

$$= [A_{G,i} * 0.8961, A_{G,i} * 1.1039]$$

To calculate the uncertainty in emissions from rice paddy areas, the upper and lower bound of the area can be incorporated into the following equation:

$$E(t) = \sum_i A_{G,i}(t) * EF_{R,i} * p_i$$

where $E(t)$ is the total emissions from rice cultivation, $p$ is the rice growing period, $EF_{R,i}$ is the emission factor at region $R$. Based on this approach, we recalculated the uncertainties for the rice cultivation sector and subsequently for the total anthropogenic emissions. All relevant results in the manuscript have been reviewed and updated accordingly.